# Genome-wide analysis highlights contribution of immune system pathways to the genetic architecture of asthma

Yi Han [1,2], Qiong Jia[1,2], Pedram Shafiei Jahani[3], Benjamin P. Hurrell[3], Calvin Pan [4], Pin Huang[1,2], Janet Gukasyan[1,2], Nicholas C. Woodward[1,2], Eleazar Eskin[5], Frank D. Gilliland [1], Omid Akbari[3], Jaana A. Hartiala [1,2,6] & Hooman Allayee [1,2,6✉]

Asthma is a chronic and genetically complex respiratory disease that affects over 300 million people worldwide. Here, we report a genome-wide analysis for asthma using data from the UK Biobank and the Trans-National Asthma Genetic Consortium. We identify 66 previously unknown asthma loci and demonstrate that the susceptibility alleles in these regions are, either individually or as a function of cumulative genetic burden, associated with risk to a greater extent in men than women. Bioinformatics analyses prioritize candidate causal genes at 52 loci, including *CD52*, and demonstrate that asthma-associated variants are enriched in regions of open chromatin in immune cells. Lastly, we show that a murine anti-CD52 antibody mimics the immune cell-depleting effects of a clinically used human anti-CD52 antibody and reduces allergen-induced airway hyperreactivity in mice. These results further elucidate the genetic architecture of asthma and provide important insight into the immunological and sex-specific relevance of asthma-associated risk variants.

[1] Department of Preventive Medicine, Keck School of Medicine, University of Southern California, Los Angeles, CA 90033, USA. [2] Department of Biochemistry & Molecular Medicine, Keck School of Medicine, University of Southern California, Los Angeles, CA 90033, USA. [3] Department of Molecular Microbiology and Immunology, Keck School of Medicine, University of Southern California, Los Angeles, CA 90033, USA. [4] Department of Human Genetics, David Geffen School of Medicine at UCLA, Los Angeles, CA 90095, USA. [5] Department of Computer Science and Inter-Departmental Program in Bioinformatics, University of California, Los Angeles, Los Angeles, CA 90095, USA. [6]These authors contributed equally: Jaana A. Hartiala, Hooman Allayee. ✉email: hallayee@usc.edu

Asthma is a global chronic respiratory disease affecting over 300 million people and a significant cause of premature death and reduced quality of life[1]. The disease develops through repeated cycles of inflammation, often in response to environmental triggers, that lead to progressive obstruction of the airways and deterioration of respiratory function[2]. While advances have been made in understanding these complex pathogenic processes, translation of research findings to the development of novel therapies has not progressed rapidly enough to address the unmet clinical needs of many patients, particularly those with severe forms of asthma.

It is generally accepted that susceptibility to asthma is a combination of genetic predisposition, exposure to environmental factors, and their interactions. Consistent with this notion, estimates for the heritability of asthma have ranged between 35% and 95%[3], although those based on twin studies have been lower[4]. Genome-wide association studies (GWAS) have confirmed the contribution of genetic factors to risk of asthma, with over 140 susceptibility loci having been identified thus far[5–13]. However, the risk alleles, most of which are common in the population, still only explain a small fraction of the overall heritability for asthma[5–7]. This observation could be due to the use of a qualitative outcome in most genetic studies, which does not depict the spectrum of asthma heterogeneity, or the existence of additional variants with smaller effect sizes, rare susceptibility alleles, and/or interactions between genes and environmental factors. In this regard, an exome sequencing study did not provide strong evidence for rare variants playing a major role in asthma susceptibility[14]. Gene-environment (GxE) interactions also remain poorly understood due to the inherent difficulties of carrying out such studies in humans, particularly with respect to accurate exposure assessment, adequately powered sample sizes with both genetic and exposure data, and the heterogeneous nature of asthma itself[15–17].

To further elucidate the genetic architecture of asthma, we carry out a GWAS analysis for asthma in the UK Biobank, followed by a meta-analysis with GWAS results from the Trans-National Asthma Genetic Consortium (TAGC)[18]. Our analyses reveal 66 regions not previously known to be associated with asthma and demonstrate that the susceptibility alleles at these loci, either individually or as a function of cumulative genetic burden, increase asthma risk to a greater extent in men than women. We also demonstrate enrichment of asthma-associated variants in regions of open chromatin in immune cells and identify candidate causal genes at 52 loci, including *CD52*. Lastly, we show that a murine anti-CD52 (α-CD52) antibody mimics the immune cell-depleting effects of a clinically used human clonal α-CD52 antibody and reduces allergen-induced airway hyperreactivity (AHR) in mice. These results expand our understanding of the genetic basis of asthma and provide evidence that sex and immune system pathways are important components of disease susceptibility.

## Results

**GWAS for asthma in the UK Biobank**. To further elucidate the genetic architecture of asthma, we first carried out a genome-wide association study (GWAS) in the UK Biobank, followed by a meta-analysis with publicly available summary data from the TAGC[18] (Fig. 1 and Supplementary Table 1 and Supplementary Data 1). The GWAS analysis in the UK Biobank included 64,538 cases and 329,321 controls and identified 32,813 significantly associated SNPs distributed among 145 loci (Supplementary Fig. 1 and Supplementary Data 2). Of these regions, 41 loci were not previously known to be associated with asthma or the combined phenotype of asthma and allergic diseases. The remaining

104 overlapped with the 146 loci previously identified for either asthma alone in adults, children, or subjects from multiple ancestries, or the combined phenotype of asthma, eczema, and hay fever[5–13] (Supplementary Data 2).

**Meta-analysis for asthma in the UK Biobank and TAGC**. We next harmonized the summary GWAS results from the TAGC to match the data in the UK Biobank (see Methods for additional details) and performed a Z-score meta-analysis for asthma with 8,365,359 SNPs common to both datasets and a total of 88,486 cases and 447,859 controls (Supplementary Table 1 and Fig. 1). The meta-analysis revealed 33,017 variants distributed across 167 loci that were associated with asthma at a genome-wide significance threshold of $P = 5.0 \times 10^{-8}$ (Fig. 2, Supplementary Fig. 2, and Supplementary Data 3). Of the 167 loci, 58 were not previously known to be associated with asthma (Fig. 2 and Table 1) whereas the other 109 overlapped with the 146 previously known loci (Supplementary Data 3). However, despite the P-values not exceeding the genome-wide significance level ($P = 5.0 \times 10^{-8}$), 34 of the remaining 37 known asthma loci did yield evidence for association in the meta-analysis at a Bonferroni-corrected threshold for testing 146 regions ($P = 0.05/146 = 3.4 \times 10^{-4}$) (Supplementary Data 4). The only exceptions were two loci for childhood asthma and one for the combined phenotype of asthma and allergic diseases (Supplementary Data 4). Eight other regions were also significantly associated with asthma in the UK Biobank alone, but their association signals fell slightly below the genome-wide significance threshold in the meta-analysis (Fig. 2 and Table 1). Thus, our collective analyses with the UK Biobank and the TAGC identified 66 loci not previously known to be associated with asthma (Fig. 2, Table 1, and Supplementary Fig. 3) and replicated nearly all (143/146) previously known genomic regions (Supplementary Data 4). Of the 66 loci, 32 were also associated with asthma in either the meta-analysis with the UK Biobank and TAGC or in the UK Biobank GWAS alone at a more stringent significance threshold of $P = 5.0 \times 10^{-9}$ (Supplementary Data 3). Altogether, the 66 loci explained an additional 1.5% of the heritability for asthma and bring the total number of susceptibility loci to 212 at the time of this analysis. However, we note that while this manuscript was under consideration, 13 of the 66 loci identified herein were reported in two other independent studies[12,13] (Table 1).

As a sensitivity analysis, we also imputed GWAS summary statistics in the TAGC using only subjects of European ancestry (19,954 asthma cases, 107,715 controls), which were then meta-analyzed with the GWAS results from the UK Biobank. This analysis revealed 51 genome-wide significant loci not previously known to be associated with asthma, of which 48 overlapped with the 58 regions identified in the multi-ethnic meta-analysis using GWAS results from TAGC that included all subjects (Supplementary Data 5). The other three loci were highly suggestive associated with asthma in the multi-ethnic meta-analysis but only achieved genome-wide significance in the meta-analysis that included subjects of European ancestry from the TAGC. Conversely, the 10 other loci that were genome-wide significant in either the UK Biobank alone or in the multi-ethnic meta-analysis with TAGC and the UK Biobank still yielded suggestive evidence for significance in the European-only meta-analysis (Supplementary Data 5). We also investigated whether there were secondary association signals at any of the 66 asthma loci, which revealed 12 independently associated SNPs at 10 loci (Supplementary Table 2). Lastly, since effect sizes were provided in the publicly available data from the TAGC[18], we also conducted a HapMap-based fixed-effects meta-analysis for asthma with 1,978,494 SNPs common to both datasets and the same number

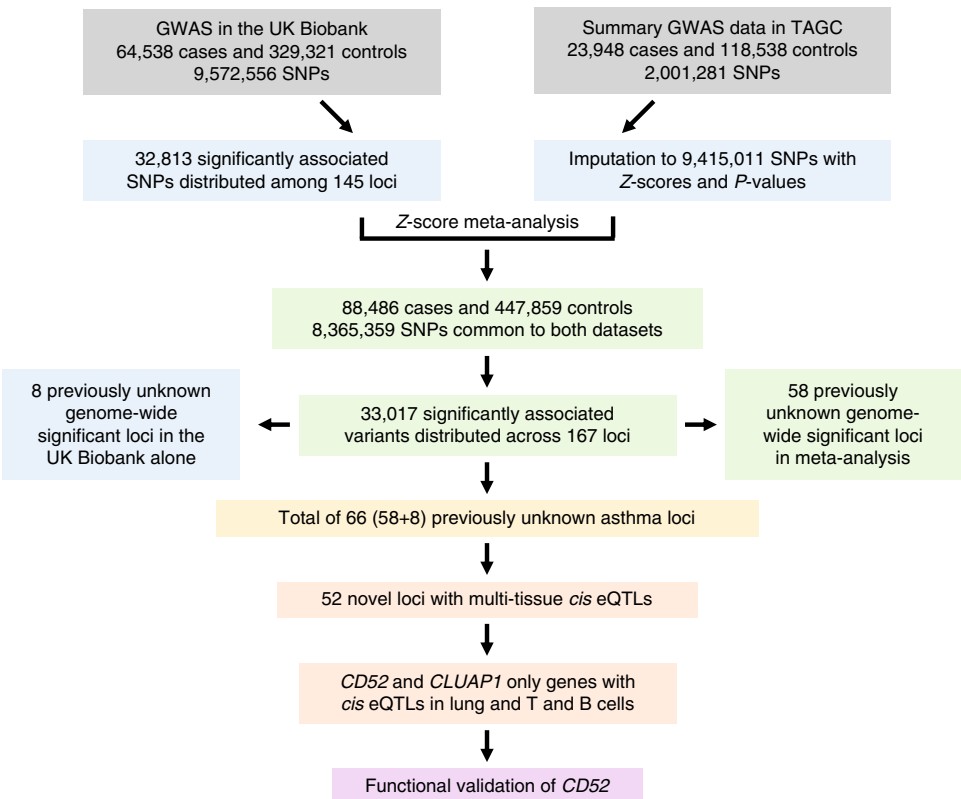

**Fig. 1 Overview of genetic and functional analyses.** A GWAS was first carried out using primary level data in the UK Biobank with ~9.5 million SNPs. In parallel, summary statistics were imputed based on publicly available GWAS data from the TAGC. The results were combined in a Z-score meta-analysis that included 88,486 asthma cases and 447,859 controls with 8,365,359 variants common to both datasets. 58 previously unknown genome-wide significant loci for asthma were identified in the meta-analysis. Combined with the eight previously unknown genome-wide significant loci from the GWAS analysis in the UK Biobank alone, a total of 66 previously unknown asthma susceptibility loci were identified. Follow up bioinformatics and eQTL analyses prioritized *CD52* for in vivo functional validation studies.

of 536,345 cases and controls. However, no additional loci were identified beyond those from the Z-score-based meta-analysis described above.

**Sex-stratified and genetic risk score analyses.** We next used primary level data from the UK Biobank to explore how association signals at the 212 asthma susceptibility loci differed in males and females and as a function of cumulative genetic burden. Thirty SNPs yielded nominal *P*-values for interaction with sex (*P*-int) that were <0.05, of which only four previously known loci on chromosomes 1p34.3, 2q37.3, 5q22.1, and 6p25.3 would be considered significant at the Bonferroni-corrected threshold for testing 212 variants ($P\text{-int} = 0.05/212 = 2.4 \times 10^{-4}$) (Supplementary Data 6). Interestingly, the effect sizes at these four loci (and most of the others) were stronger in males than in females, although it should be noted that the significance of the interaction with the chromosome 1p34.3 locus could be inflated due to frequency of the susceptibility allele (Supplementary Data 6).

We next constructed three weighted genetic risk scores (GRS) in the UK Biobank based on the number of risk alleles carried at the 66 previously unknown, 146 known, or all 212 loci. Each GRS was weighted by the overall effect sizes of the included alleles that were derived from the association analysis that included both sexes. Individuals in the highest decile for the GRS constructed with all 212 risk alleles had nearly a 5-fold increased risk of asthma compared to those in the lowest decile (OR = 4.7, 95% CI 4.5–4.9; $P < 1.0 \times 10^{-305}$) (Fig. 3a), and a 2.5-fold increased risk of

asthma compared to those in the middle decile (OR = 2.5, 95% CI 2.4–2.6; $P < 1.0 \times 10^{-305}$). Increased risk of asthma was also observed for individuals in the highest versus lowest deciles of the GRS for the 146 known and 66 previously unknown loci (Fig. 3a), although, as expected, the overall effect size for the 66-locus GRS was weaker (OR = 2.0, 95% CI 1.9–2.1; $P = 4.3 \times 10^{-266}$).

To determine whether cumulative genetic burden differed in men and women, we also used sex-specific effect sizes to construct a weighted GRS in men and women separately for the 66 previously unknown, 146 known, or all 212 asthma loci. This analysis revealed a pattern where increased risk of asthma among subjects in the highest versus lowest decile of the GRS was approximately one order of magnitude greater in men than in women (Fig. 3b–d). For example, compared to the lowest decile, the OR for asthma among men in the highest decile for the GRS constructed with all 212 loci was 5.5 (95% CI 5.2–5.9; $P < 1.0 \times 10^{-305}$) whereas the same analysis in women revealed that increased risk of asthma among subjects in the highest decile was 4.4 (95% CI 4.2–4.6; $P < 1.0 \times 10^{-305}$). These results were accompanied by significant gene-sex interactions when comparing the highest versus lowest decile for the GRS constructed from all 212 ($P\text{-int} = 7.3 \times 10^{-10}$), 146 known ($P\text{-int} = 1.9 \times 10^{-10}$), and 66 previously unknown ($P\text{-int} = 0.03$) loci. Significant interactions with sex were also observed when considering the entire spectrum of cumulative genetic burden for the 212 ($P\text{-int} = 6.1 \times 10^{-18}$), 146 known ($P\text{-int} = 3.0 \times 10^{-18}$), and 66 previously unknown ($P\text{-int} = 0.01$) loci. The interactions with sex across all deciles remained significant even after exclusion of the four risk alleles on

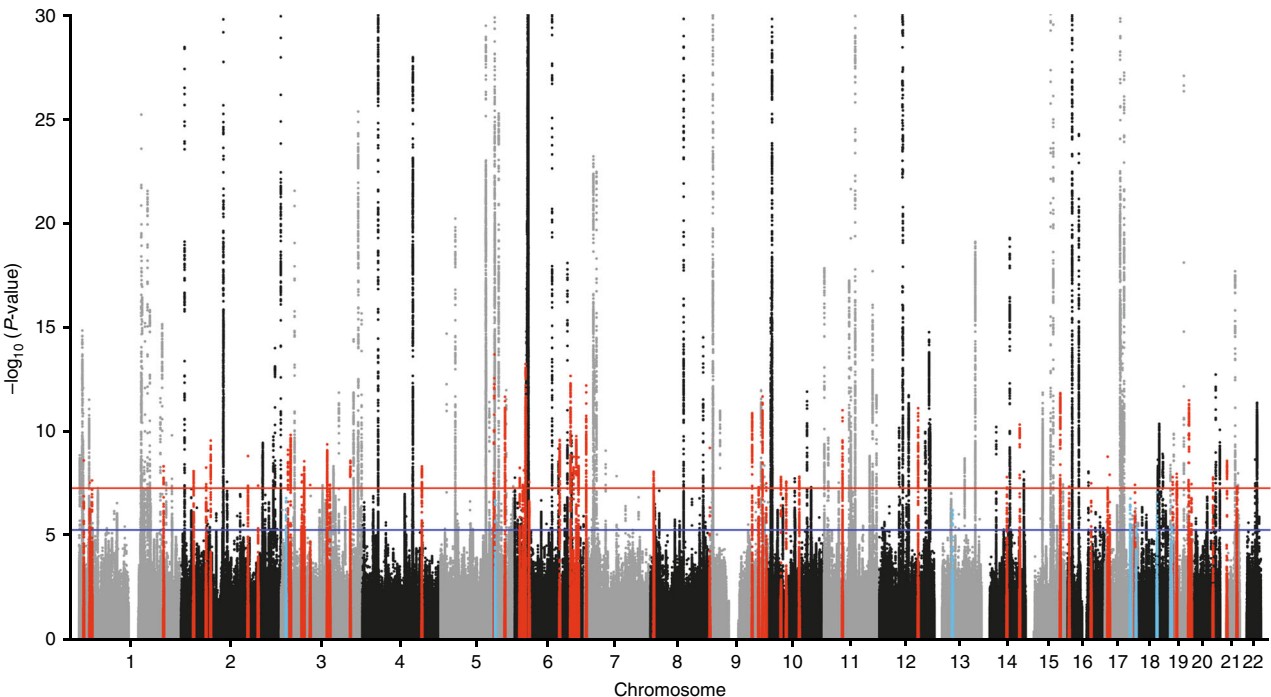

**Fig. 2 GWAS meta-analysis results for asthma.** Of the 66 previously unknown loci, 58 were significantly associated with asthma in the meta-analysis with the UK Biobank and TAGC (red dots). Eight loci were associated with asthma in the GWAS analysis with the UK Biobank alone but fell slightly below the genome-wide significance threshold in the meta-analysis with the TAGC (light blue dots). The Z-score meta-analysis included a total of 88,486 cases and 447,859 controls from the UK Biobank (64,538 asthma cases and 329,321 controls) and TAGC (23,948 asthma cases and 118,538 controls) and 8,365,359 SNPs common to both datasets. Genome-wide thresholds for significant ($P = 5.0 \times 10^{-8}$) and suggestive ($P = 5.0 \times 10^{-6}$) association are indicated by the horizontal red and dark blue lines, respectively. $P$-values are truncated at $-\log_{10}(P) = 30$.

chromosomes 1p34.3, 2q37.3, 5q22.1, and 6p25.3 from the 212- ($P$-int $= 5.1 \times 10^{-14}$) and 146-locus ($P$-int $= 5.5 \times 10^{-14}$) GRS.

**Associations with asthma-related phenotypes.** We next carried out sensitivity analyses in the UK Biobank to compare the effect sizes of the association signals at the 66 loci for various definitions of asthma and allergic diseases only (hay fever, eczema, allergy, or allergy to house dust mite) (Supplementary Data 7). Compared to association signals for allergic diseases only, most of the 66 loci were more strongly associated with broadly defined asthma, where subjects with allergic diseases were still included as a case as long as they had a positive diagnosis of asthma. Similarly, association signals were also stronger with more strict definitions of asthma that excluded subjects who either had allergic diseases or chronic obstructive pulmonary disease (COPD), emphysema, and chronic bronchitis (Supplementary Data 7). In addition, association signals at some of the 66 loci were stronger with the two strict definitions of asthma than with broadly defined asthma (Supplementary Data 7). Restricting the analysis to only subjects of European ancestry in the UK Biobank yielded similar findings (Supplementary Data 7).

We next evaluated the 66 asthma loci for associations with lung function traits. At the Bonferroni-corrected significance threshold for testing 66 loci ($P = 0.05/66 = 7.6 \times 10^{-4}$), the asthma risk alleles at 24 of the loci exhibited directionally consistent associations with decreased forced expiratory volume in one second (FEV$_1$), forced vital capacity (FVC), the ratio of FEV$_1$/FVC, and peak expiratory flow (PEF) (Supplementary Data 8). To further explore their clinical relevance, we used the PhenoScanner database[19] to determine whether the loci identified in our meta-analyses were associated with other disease phenotypes. Interestingly, the lead variants (or tightly linked proxies) at

approximately half of the loci had been previously associated with other asthma-relevant traits, including blood cell parameters (19 loci), height (7 loci), BMI and waist circumference (4 loci), inflammatory bowel disease (3 loci), and cardiovascular disease (1 locus) (Supplementary Data 9).

**Enrichment analyses with asthma-associated variants.** To gain biological insight into the previously unknown asthma loci, we used two bioinformatics tools to carry out enrichment analyses for regulatory elements and pathways. GWAS analysis of regulatory or functional information enrichment with LD correction (GARFIELD) revealed highly significant 3 to 4-fold enrichment of asthma-associated variants colocalizing to DNase I hypersensitive sites in several tissues with biological relevance to asthma, such as epithelium, lung, and thymus (Fig. 4a; Supplementary Data 10). However, enrichment in DNase I hypersensitive elements was even more evident in blood, and specifically in GM12878 B cells, CD20$^+$ B cells, and CD3$^+$, CD4$^+$, or CD8$^+$ T cells (Fig. 4b; Supplementary Data 10). A Data-driven Expression-Prioritized Integration for Complex Trait (DEPICT) analysis also yielded consistent results where the most significantly enriched biological pathways and tissues were leukocyte activation, differentiation, or proliferation, cytokine production, lymphoid organ development, blood cells, lymphocytes, lymphoid tissue, and spleen (Supplementary Data 11, 12). Taken together, these results further highlight immune system pathways as important components of the pathophysiology of asthma.

**Prioritization of loci and positional candidate genes.** We next used expression quantitative trait locus (eQTL) data from various publicly available sources, such as the GTEx Project and eQTL-Gen Consortium (Supplementary Data 13), to prioritize

**Table 1 66 loci identified for asthma in meta-analysis of the UK Biobank and TAGC.**

| SNP | CHR | BP (hg19) | Nearest Gene(s) | EA/NEA | EAF | P_UKBB | P_TAGC | P_Meta-analysis | Direction | P-het |
|---|---|---|---|---|---|---|---|---|---|---|
| rs2230624[a,b] | 1p36.22 | 12,175,658 | TNFRSF8 | G/A | 0.99 | 4.3E−12 | 0.95 | 2.4E−09 | ++ | 4.4E−04 |
| rs11577318 | 1p36.11 | 26,601,570 | CEP85 | G/A | 0.18 | 4.8E−05 | 5.8E−05 | 2.8E−09 | ++ | 0.18 |
| rs2228552[b] | 1p35.2 | 32,165,495 | COL16A1 | T/G | 0.63 | 3.4E−08 | 0.09 | 2.2E−08 | ++ | 0.16 |
| rs4127124 | 1q32.1 | 206,624,362 | SRGAP2 | A/C | 0.49 | 4.6E−09 | 0.10 | 4.5E−09 | ++ | 0.11 |
| rs72787718 | 2p23.1 | 30,474,351 | LBH | A/G | 0.23 | 3.2E−08 | 0.04 | 7.7E−09 | ++ | 0.26 |
| rs68110799 | 2p16.1 | 60,945,555 | PAPOLG | A/G | 0.26 | 8.9E−08 | 0.01 | 5.2E−09 | ++ | 0.50 |
| rs35548551 | 2p13.2 | 72,039,714 | DYSF | T/G | 0.12 | 4.4E−07 | 1.1E−04 | 2.7E−10 | ++ | 0.48 |
| rs6741949 | 2q24.2 | 162,910,223 | DPP4 | G/C | 0.56 | 1.0E−08 | 0.03 | 1.5E−09 | ++ | 0.29 |
| rs2595389 | 2q32.1 | 187,534,183 | ITGAV | T/G | 0.54 | 1.0E−05 | 8.9E−04 | 3.8E−08 | ++ | 0.57 |
| rs13101202[c] | 3p25.2 | 12,699,361 | RAF1 | C/A | 0.17 | 2.1E−09 | 0.81 | 1.5E−07 | ++ | 3.9E−03 |
| rs2014490 | 3p24.3 | 16,972,211 | PLCL2 | G/A | 0.65 | 9.2E−10 | 0.08 | 7.5E−10 | ++ | 0.10 |
| rs2036226 | 3p24.3 | 23,597,986 | UBE2E2 | T/C | 0.59 | 7.5E−08 | 4.7E−04 | 1.4E−10 | ++ | 0.82 |
| rs73072483 | 3p21.2 | 50,771,624 | DOCK3 | G/A | 0.87 | 1.4E−07 | 0.02 | 1.2E−08 | ++ | 0.47 |
| rs9813229 | 3p14.3 | 56,741,585 | ARHGEF3 | A/G | 0.62 | 1.4E−07 | 5.1E−03 | 2.6E−09 | ++ | 0.75 |
| rs7372960 | 3p13 | 71,254,751 | FOXP1 | T/C | 0.75 | 1.4E−07 | 0.05 | 3.5E−08 | ++ | 0.29 |
| rs7622814 | 3q13.2 | 112,650,431 | CD200R1 | T/G | 0.46 | 9.8E−09 | 9.5E−03 | 4.1E−10 | ++ | 0.46 |
| rs9877891 | 3q13.33 | 119,260,866 | CD80 | C/T | 0.18 | 2.7E−08 | 0.14 | 3.4E−08 | ++ | 0.11 |
| rs17485347 | 3q26.2 | 169,127,519 | MECOM | C/A | 0.75 | 2.3E−08 | 2.0E−04 | 2.5E−09 | ++ | 0.45 |
| rs6842889 | 4q31.21 | 145,479,880 | LOC105377462 | T/C | 0.61 | 3.0E−06 | 3.1E−04 | 4.6E−09 | ++ | 0.49 |
| rs11950815 | 5q31.1 | 130,955,487 | RAPGEF6 | G/A | 0.63 | 8.6E−12 | 4.8E−04 | 2.0E−14 | ++ | 0.60 |
| rs62379371[c] | 5q31.1 | 133,439,274 | LOC105379185 | G/A | 0.95 | 6.5E−09 | 0.79 | 3.2E−07 | ++ | 0.01 |
| rs11746314[a,b] | 5q33.3 | 156,752,957 | CYFIP2 | G/A | 0.06 | 4.2E−11 | 7.7E−03 | 2.1E−12 | ++ | 0.26 |
| rs3777755 | 6p24.1 | 12,159,699 | HIVEP1 | C/T | 0.70 | 2.1E−07 | 7.0E−03 | 5.3E−09 | ++ | 0.72 |
| rs13203384 | 6p22.3 | 19,341,301 | MBOAT1 | A/G | 0.21 | 1.8E−06 | 5.8E−03 | 3.5E−08 | ++ | 0.92 |
| rs952579 | 6p22.3 | 21,884,440 | CASC15 | A/G | 0.14 | 2.4E−09 | 0.19 | 7.1E−09 | ++ | 0.05 |
| rs766406[a] | 6p22.2 | 26,319,588 | HIST1H4H/BTN3A2 | T/G | 0.65 | 7.8E−09 | 6.4E−07 | 5.8E−14 | ++ | 0.20 |
| rs9394288 | 6p21.31 | 35,222,010 | LOC101929285 | T/C | 0.20 | 2.3E−09 | 0.38 | 2.6E−08 | ++ | 0.02 |
| rs11759732[a] | 6q21 | 109,370,006 | SESN1 | A/G | 0.85 | 1.9E−08 | 3.4E−03 | 2.5E−10 | ++ | 0.70 |
| rs4526212[a] | 6q23.3 | 135,804,631 | AHI1 | A/C | 0.37 | 6.5E−11 | 7.2E−04 | 2.1E−13 | ++ | 0.64 |
| rs13190880 | 6q24.2 | 143,222,770 | HIVEP2 | T/G | 0.09 | 8.5E−08 | 5.1E−03 | 1.6E−09 | ++ | 0.72 |
| rs431362 | 6q25.1 | 149,787,378 | ZC3H12D | A/G | 0.36 | 1.5E−08 | 2.8E−03 | 1.6E−10 | ++ | 0.72 |
| rs4599658 | 6q25.2 | 155,074,505 | SCAF8 | G/T | 0.47 | 1.8E−08 | 0.02 | 1.7E−09 | ++ | 0.37 |
| rs73033536[a] | 7p22.2 | 3,149,883 | LOC105375130 | T/C | 0.91 | 8.1E−11 | 1.6E−03 | 6.2E−13 | ++ | 0.52 |
| rs7014953 | 8p23.1 | 8,168,413 | SGK223 | A/C | 0.60 | 2.3E−07 | 9.9E−03 | 8.3E−09 | ++ | 0.65 |
| rs34173062[b] | 8q24.3 | 145,158,607 | MAF1/SHARPIN | A/G | 0.07 | 1.3E−12 | 0.83 | 6.0E−10 | ++ | 5.2E−04 |
| rs1537504[a,b] | 9q22.33 | 101,829,542 | COL15A1 | A/G | 0.23 | 1.5E−07 | 1.1E−05 | 1.3E−11 | ++ | 0.29 |
| rs4978607 | 9q32 | 117,508,437 | TNFSF15 | C/T | 0.88 | 2.3E−08 | 0.19 | 4.5E−08 | ++ | 0.08 |
| rs10986311 | 9q33.3 | 127,071,493 | NEK6 | C/T | 0.37 | 2.7E−08 | 1.1E−05 | 2.1E−12 | ++ | 0.37 |
| rs782134971 | 9q34.2 | 136,139,907 | ABO | -/AAACTGCC | 0.25 | 1.5E−05 | 3.7E−04 | 3.0E−08 | ++ | 0.41 |
| rs9284092 | 10p11.23 | 30,801,718 | MAP3K8 | G/T | 0.73 | 8.5E−06 | 3.3E−04 | 1.5E−08 | ++ | 0.43 |
| rs12769745 | 10q11.21 | 43,749,700 | RASGEF1A | A/G | 0.28 | 1.5E−06 | 4.8E−03 | 2.4E−08 | ++ | 0.95 |
| rs1134777 | 10q22.2 | 75,538,651 | FUT11 | C/G | 0.73 | 2.3E−07 | 0.02 | 1.4E−08 | ++ | 0.54 |
| rs714417 | 11p11.2 | 45,247,176 | PRDM11 | C/T | 0.69 | 7.5E−08 | 1.9E−05 | 9.5E−12 | ++ | 0.37 |
| rs12303699[a,b] | 12q22 | 94,582,336 | PLXNC1 | A/G | 0.38 | 1.9E−09 | 9.7E−04 | 7.5E−12 | ++ | 0.79 |
| rs61960013[a,c] | 13q14.11 | 44,490,181 | LOC107984576 | G/C | 0.21 | 7.0E−09 | 0.96 | 6.1E−07 | ++ | 3.3E−03 |
| rs3751289 | 14q23.1 | 61,983,943 | PRKCH | G/A | 0.22 | 9.8E−08 | 0.03 | 1.5E−08 | ++ | 0.35 |
| rs10131197 | 14q32.12 | 93,015,394 | RIN3 | G/A | 0.33 | 2.6E−10 | 0.02 | 4.6E−11 | ++ | 0.19 |
| rs11259930[a] | 15q25.2 | 84,577,350 | ADAMTSL3 | A/G | 0.47 | 8.5E−09 | 3.1E−05 | 1.4E−12 | ++ | 0.55 |
| rs11645975 | 16p13.3 | 3,749,397 | TRAP1 | G/C | 0.74 | 3.5E−05 | 1.7E−04 | 4.2E−08 | ++ | 0.28 |
| rs223819 | 16q13 | 57,394,862 | CCL22 | T/C | 0.91 | 2.3E−06 | 3.7E−03 | 3.0E−08 | ++ | 0.96 |
| rs72842819 | 17p13.1 | 7,328,821 | C17orf74/SPEM2 | C/A | 0.12 | 3.2E−09 | 0.06 | 1.6E−09 | ++ | 0.15 |
| rs750065349 | 17p12 | 12,193,443 | MAP2K4 | GC/G | 0.46 | 8.6E−10 | 0.38 | 1.1E−08 | ++ | 0.02 |
| rs1991401[c] | 17q23.3 | 62,502,435 | CEP95/DDX5 | A/G | 0.69 | 1.1E−08 | 0.50 | 1.6E−07 | ++ | 0.02 |
| rs111365807 | 17q25.1 | 73,825,463 | UNC13D | C/G | 0.12 | 1.2E−07 | 0.06 | 3.5E−08 | ++ | 0.27 |
| rs76848919[c] | 17q25.3 | 76,352,554 | SOCS3 | C/G | 0.20 | 5.6E−09 | 0.55 | 2.7E−06 | ++ | 4.5E−04 |
| rs12956924[c] | 18q21.1 | 46,451,146 | SMAD7 | A/G | 0.32 | 3.7E−08 | 0.21 | 7.9E−08 | ++ | 0.08 |
| rs117552144[b,c] | 19p13.3 | 3,136,091 | GNA15 | T/C | 0.06 | 3.6E−11 | 0.26 | 3.5E−07 | ++ | 1.2E−05 |
| rs10420217[c] | 19p13.3 | 4,355,871 | MPND | C/T | 0.28 | 2.0E−08 | 0.21 | 3.1E−05 | ++ | 7.6E−05 |
| rs755023315 | 19p13.3 | 6,579,029 | CD70 | G/GC | 0.29 | 2.7E−09 | 0.27 | 1.5E−08 | ++ | 0.03 |
| rs34006614 | 19p13.11 | 16,442,782 | KLF2 | T/C | 0.33 | 1.1E−07 | 0.02 | 1.0E−08 | ++ | 0.43 |
| rs143432496 | 19q13.32 | 45,252,714 | BCL3 | A/ATAT | 0.27 | 2.3E−11 | 0.10 | 4.8E−11 | ++ | 0.04 |
| rs8103278[a,b] | 19q13.32 | 46,370,381 | FOXA3 | G/A | 0.65 | 1.2E−08 | 5.1E−05 | 3.1E−12 | ++ | 0.59 |
| rs11670020 | 19q13.41 | 52,314,161 | FPR3/LOC105369197 | G/A | 0.14 | 1.7E−06 | 5.6E−03 | 3.2E−08 | ++ | 0.93 |
| rs8125525 | 20q13.12 | 45,681,788 | EYA2 | C/T | 0.74 | 1.7E−07 | 0.02 | 1.5E−08 | ++ | 0.46 |
| rs1736147 | 21q21.1 | 16,813,053 | LOC101927745 | G/A | 0.57 | 1.1E−07 | 6.1E−03 | 2.5E−09 | ++ | 0.70 |
| rs34846236 | 21q22.2 | 41,049,344 | PCP4 | G/GC | 0.48 | 5.3E−06 | 2.2E−03 | 4.3E−08 | ++ | 0.78 |

*EA/NEA* effect (asthma risk) allele/non-effect allele, *EAF* effect allele frequency (based on only the UK Biobank).
*P-UKBB* p-value from the GWAS in the UK Biobank as calculated by linear mixed models, *P_TAGC* imputed p-value from TAGC GWAS as calculated by SSimp software[63], *P-het* p-value for test of heterogeneity between the UK Biobank and TAGC.
[a]Identified by Johansson et al.[12] while this manuscript was under consideration.
[b]Identified by Olafsdottir et al.[13] while this manuscript was under consideration.
[c]Locus was genome-wide significant (P < 5.0E−08) in the UK Biobank alone.

candidate causal genes at the 66 previously unknown regions (Fig. 1). The lead SNP or tightly linked ($r^2 > 0.8$) proxy variants at 52 loci yielded at least one *cis* eQTL for a positional candidate gene in one or more tissues, including those relevant to asthma (Supplementary Data 13). Notably, the most significant eQTLs were in blood, some of which yielded $P < 1.0 \times 10^{-300}$. These

observations are consistent with the GARFIELD enrichment analyses pointing to immune cells in blood as playing an important role in asthma.

To prioritize loci for functional validation, we focused on positional candidate genes for which eQTLs were observed in immune cell populations that were highlighted by our enrichment

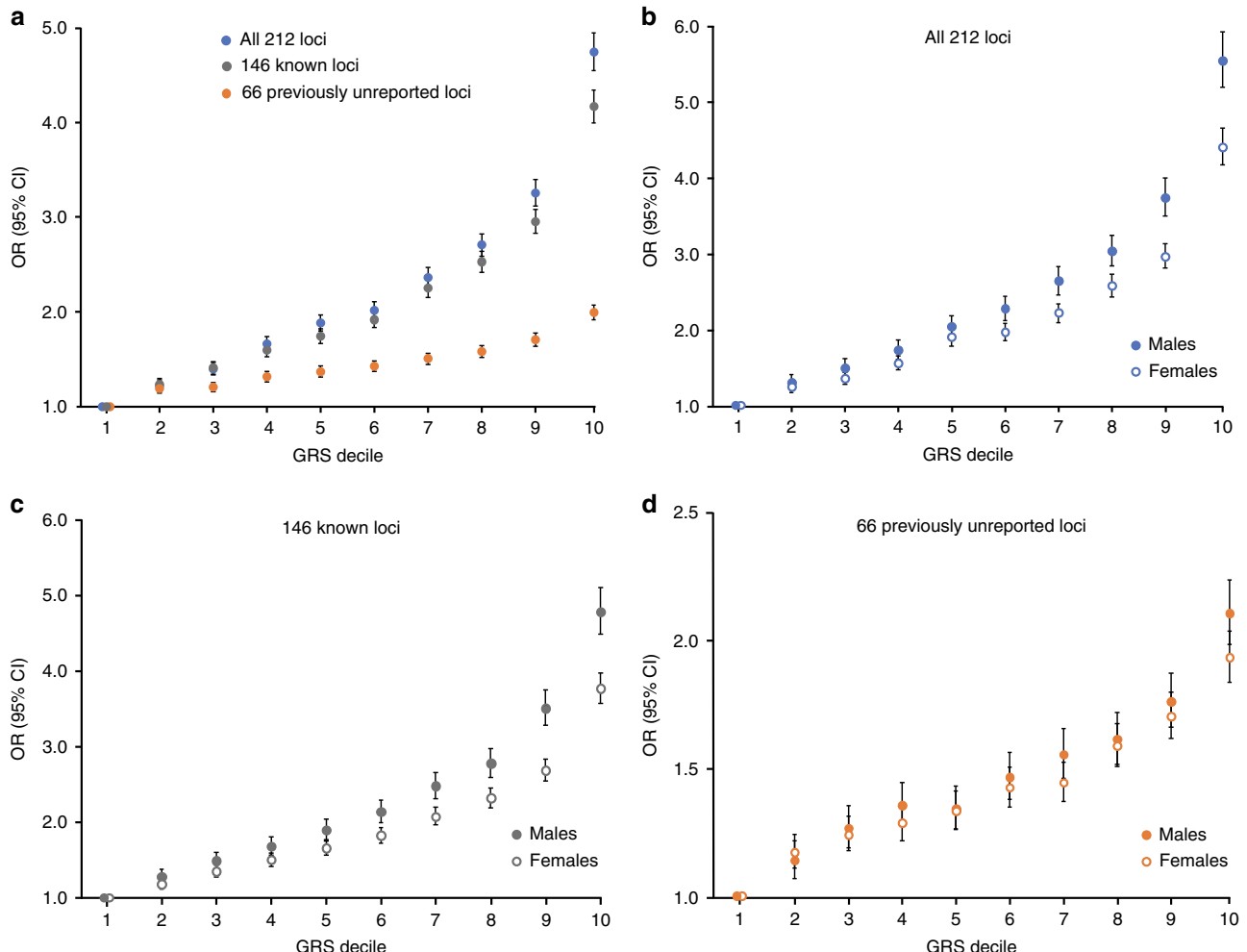

**Fig. 3 Cumulative genetic burden and risk of asthma in the UK Biobank. a** Individuals in the highest decile for three weighted genetic risk scores (GRS) based on the known or previously unreported susceptibility loci have significantly elevated risk for asthma compared to the lowest decile. **b–d** Compared to females, cumulative genetic risk for asthma was more significantly pronounced in males across the entire spectrum of sex-specific weighted GRS constructed from all 212 (P-int = $6.1 \times 10^{-18}$), 146 known (P-int = $3.0 \times 10^{-18}$), and 66 previously unknown (P-int = 0.01) loci. Interactions with sex were also significant when comparing the highest versus lowest decile for the GRS constructed from all 212 (P-int = $7.3 \times 10^{-10}$), 146 known (P-int = $1.9 \times 10^{-10}$), and 66 previously unknown (P-int = 0.03) loci. Each decile in the GRS analyses included 39,386 subjects when both sexes were combined (**a**) or 18,318 males and 21,068 females when stratified by sex (**b–d**). Data are shown as ORs with 95% CIs, as calculated by logistic regression.

analyses, such as B and T cells. *CD52*, *AHI1*, and *CLUAP1* on chromosomes 1p36.11, 6q23.3, and 16p13.3 (Supplementary Fig. 3), respectively, were three genes that met these criteria and yielded the most significant eQTLs in these specific immune cell types (Supplementary Data 13). However, eQTLs were also observed in the lung for *CD52* and *CLUAP1*, making these even stronger candidate causal genes for functional validation (Supplementary Data 13). *CLUAP1* was initially identified as a protein that interacts with clusterin and is required for ciliogenesis[20] but has not otherwise been directly implicated in asthma or pulmonary biology. By comparison, *CD52* encodes a membrane glycoprotein present at varying levels on the surface of various leukocytes but not hematopoietic cells[21]. In addition, alemtuzumab is a monoclonal anti-CD52 (αCD52) antibody that results in preferential and prolonged depletion of circulating T and B cells[22,23] and is FDA approved for the treatment of B-cell chronic lymphocytic leukemia[24–26] and relapsing remitting multiple sclerosis[27–29]. Because of its biological role in immune cells relevant to asthma as well as its potential translational implications, we, therefore, prioritized *CD52* as a strong causal positional candidate gene for functional validation (Fig. 1).

**Functional validation of CD52**. To functionally validate *CD52*, we designed an in vivo pharmacological experiment in mice that would mimic the clinical protocol for administering alemtuzumab to humans, in conjunction with the induction of AHR through exposure to house dust mite (HDM) (Supplementary Fig. 4a and Fig. 5a). However, since alemtuzumab does not target the mouse CD52 protein, we used flow cytometry (Supplementary Figs. 4b–d) to first investigate the immune cell depleting activity of a mouse αCD52 antibody that was previously reported to have biological effects[30]. Consistent with the clinical effects of alemtuzumab on circulating lymphocytes in humans[22,23], the mouse αCD52 antibody efficiently depleted CD4[+] and CD8[+] T cells as well as CD19[+] B cells in both the lung and spleen (Supplementary Fig. 5). We next tested whether the murine αCD52 antibody reduced HDM-induced AHR and lung inflammation (Fig. 5a). As expected, mice exposed to HDM had significantly increased lung resistance and decreased dynamic compliance compared to phosphate-buffered saline (PBS)-exposed mice (Fig. 5b, c). However, the induction of lung resistance and deterioration of dynamic compliance as a result of HDM exposure was significantly blunted in mice administered

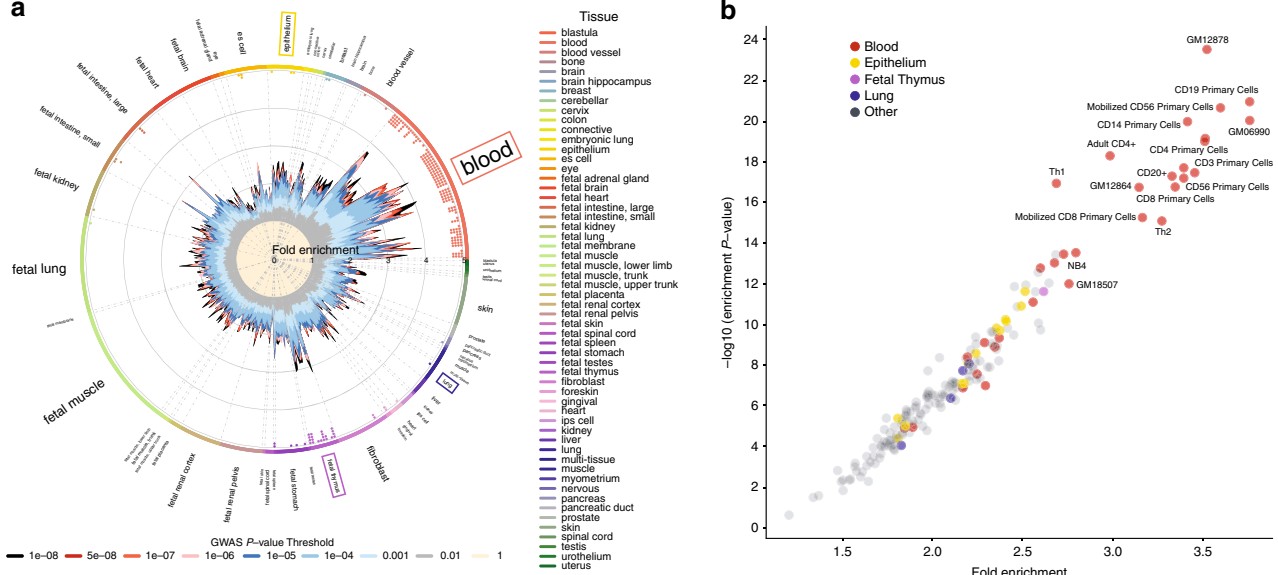

**Fig. 4 Enrichment analyses with asthma-associated variants. a** GARFIELD analysis revealed highly significant 3 to 4-fold enrichment of asthma-associated variants colocalizing to DNase I hypersensitive sites (peaks) in several asthma-related tissues, such as blood (red box), epithelium (orange box), lung (blue box), and thymus (purple box). The radial plot shows fold enrichment in various available tissues using different GWAS significance thresholds for the inclusion of asthma-associated variants in the GARFIELD analysis, as calculated by logistic regression. The small dots on the inside of the outer most circle indicate whether the inclusion of asthma-associated variants at GWAS significance thresholds of $1.0 \times 10^{-5}$ (one dot), $1.0 \times 10^{-6}$ (two dots), $1.0 \times 10^{-7}$ (three dots), $1.0 \times 10^{-8}$ (four dots) in the GARFIELD analysis yielded significant enrichment in that tissue or cell type at $P = 1.0 \times 10^{-15}$. **b** Highly significant enrichment in DNase I hypersensitive sites was particularly evident in immune cells, such as GM12878 B cells, CD19[+] B cells as well as CD3[+], CD4[+], and CD8[+] T cells.

the αCD52 antibody compared to the isotype control antibody (Fig. 5b, c). Injection of HDM-exposed mice with the αCD52 antibody also significantly decreased numbers of total CD45[+] cells, eosinophils, T cells, and neutrophils in bronchial alveolar lavage (BAL) (Fig. 5d). Consistent with these observations, histological analysis of the lung showed reduced infiltration of inflammatory cells and decreased thickness of the airway epithelium in HDM-exposed mice injected with the αCD52 antibody compared to the isotype control antibody (Fig. 5e). Collectively, these data demonstrate that targeting CD52 with an antagonizing antibody ameliorates cellular and physiological lung function traits in mice exposed to HDM and provide functional in vivo evidence that *CD52* is at least one candidate causal gene at the risk locus on chromosome 1p36.11.

## Discussion

In the present study, we identified 66 regions not previously known to be associated with asthma and replicated all but three of the 146 known susceptibility loci. Altogether, the 212 loci still only explain ~8–9% of the total heritability for asthma, of which ~1.5% can be attributed to the 66 previously unreported loci. The relatively low fraction of the heritability explained by genetic factors is likely a reflection of the susceptibility alleles' modest effect sizes on disease risk and the more complex underlying phenotypic and genetic heterogeneity of asthma compared to other common diseases[31]. For example, although the association signals were not genome-wide significant, many of the asthma loci did exhibit suggestive associations with allergic diseases as well. These findings are consistent with recent studies demonstrating that asthma and allergic diseases have shared genetic determinants[12,32]. By comparison, less than half of the 66 asthma loci exhibited associations with FEV$_1$, FVC, FEV$_1$/FVC, or PEF, consistent with the proportion that previous studies have reported[33]. These observations suggest that not all of the genetic factors

predisposing to asthma necessarily manifest through associations with pulmonary function traits.

GxE interactions are also recognized as important components of asthma's complex genetic etiology but progress in this area has been limited, with most studies focusing on exposures such as allergens, cigarette smoke, and air pollutants[15–17]. Our data suggest that sex is another potentially important endogenous factor that can modulate risk through interactions with asthma susceptibility alleles, both individually or as a function of cumulative genetic burden. This was particularly evident among men in whom increased risk of asthma across all categorized risk alleles was consistently more elevated than in females, even after the exclusion of four susceptibility alleles that individually yielded statistical evidence for being more strongly associated with asthma in men. Evidence for such gene-sex interactions has previously been demonstrated for asthma[34–37] as well as several other complex phenotypes[38], including anthropometric traits[39–41] and coronary artery disease[42]. Interestingly, the male-specific association (rs2549003) reported at the known chromosome 5q31.1 locus[36] was associated with asthma in our meta-analysis (overall $P = 2.0 \times 10^{-14}$) but the association signal was derived equally from men and women. Nonetheless, our data collectively suggest that gene-sex and other GxE interactions are likely to have important contributions to risk of asthma. Future studies will be required to explore these areas further with respect to discovery, particularly as new statistical methods are developed to carry out GxE analyses on a genome-wide level[43], as well as to understand the underlying biological mechanisms.

Our results also add to the growing body of genetic and bioinformatics evidence that immune system-related processes are important drivers of asthma pathophysiology, consistent with prior studies[18]. For example, enrichment analyses highlighted pathways related to various aspects of leukocyte function and immune system regulation/development. More specifically, asthma-associated variants were highly enriched in regions of

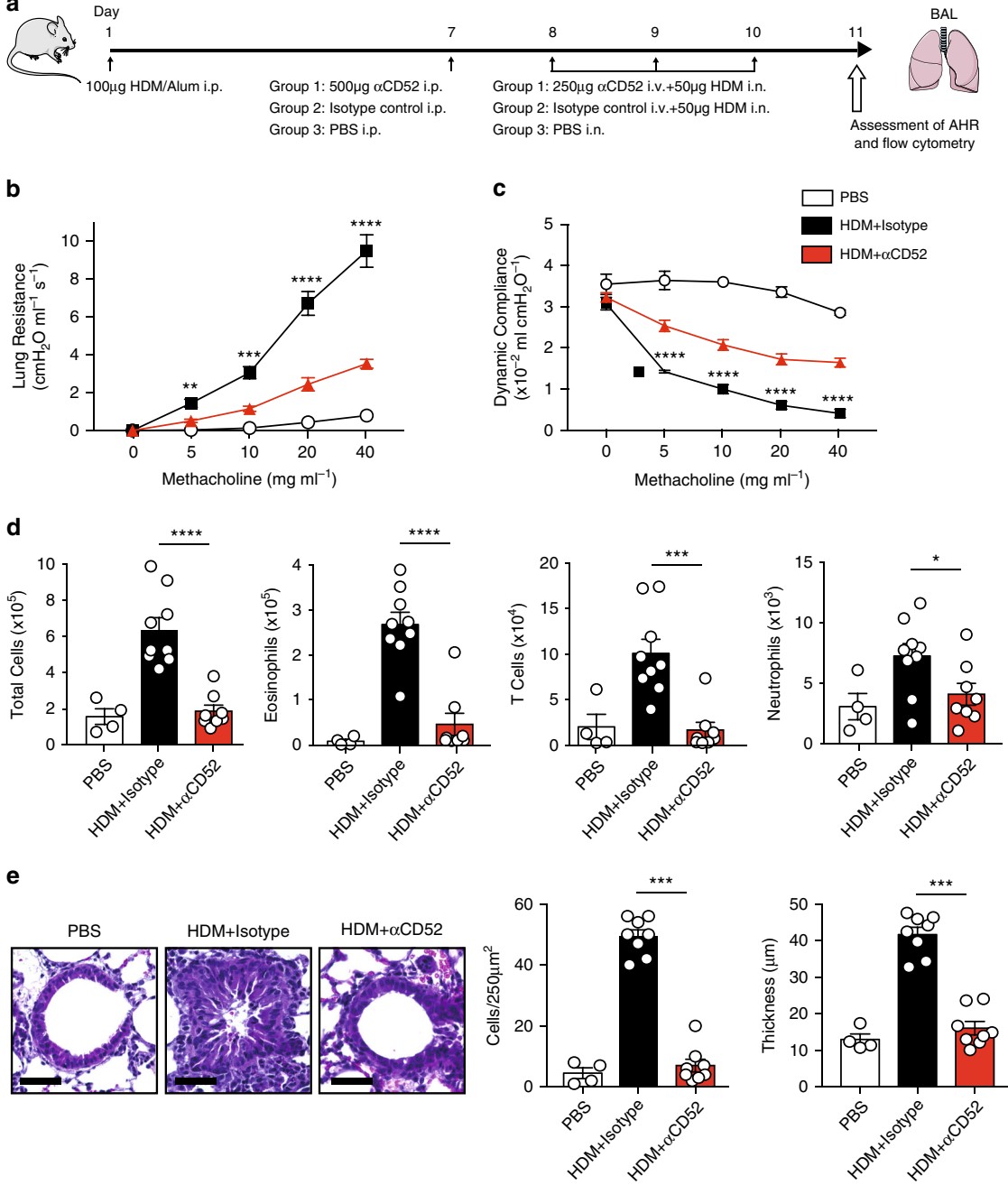

**Fig. 5 Effect of an anti-CD52 antibody on lung function and inflammation in mice. a** The experimental protocol for testing the effect of a mouse anti-CD52 (αCD52) antibody on pulmonary function and inflammation was designed to mimic the clinical protocol used to treat multiple sclerosis patients with a human αCD52 antibody (alemtuzumab). Female BALB/cByJ mice were immunized on day 1 with 100 μg of house dust mite (HDM) in 2 mg of aluminum hydroxide (alum) by intraperitoneal (i.p.) injection. On day 7, mice were intraperitoneally administered with 500 μg of either the αCD52 antibody (Group 1, red bars; n = 8) or isotype control antibody (Group 2, black bars; n = 9), or phosphate-buffered saline (PBS) (Group 3, white bars; n = 4). On days 8, 9 and 10, mice in Groups 1 and 2 were intravenously (i.v.) administered 250 μg of the αCD52 antibody or isotype control antibody, respectively, and simultaneously challenged intranasally (i.n.) with 50 μg HDM. Mice in Group 3 were only challenged with PBS i.n. on days 8, 9, and 10. On day 11, lung function was measured by invasive plethysmography and leukocyte counts were determined in bronchial alveolar lavage (BAL) by flow cytometry. Compared to HDM-exposed mice receiving the control isotype antibody, HDM-exposed mice administered the αCD52 antibody had significantly reduced lung resistance (**b**) and improved dynamic compliance (**c**). HDM-exposed mice receiving the αCD52 antibody also had significantly decreased numbers of total cells, eosinophils, T cells, and neutrophils in BAL (**d**) as well as decreased inflammation and thickness of the airway epithelium (**e**) compared to HDM-exposed mice injected with the isotype control antibody. Scale bars equal 50 μm. Data are shown as mean ± SE and derived from two independent experiments yielding similar results. Two-tailed unpaired Student's $t$-tests were used to determine statistically significant differences between groups. *$P <$ 0.05, **$P <$ 0.005; ***$P <$ 0.0005; ****$P <$ 0.0001.

open chromatin in several tissues and especially in B and T lymphocytes. This concept is also supported by the nature of specific candidate causal genes located at some of the susceptibility loci. One such example is the chromosome 1p36.22 locus, which was identified independently in two other recent studies while this manuscript was under consideration[12,13]. In our analysis, the lead SNP (rs2230624; G>A) yielded the strongest effect size for asthma (OR = 1.19) out of all 212 risk variants. Rs2230624 leads to a rare (~1% frequency in European populations) but computationally predicted deleterious Cys273Tyr substitution in exon 8 of *TNFRSF8*, which encodes the costimulatory molecule CD30. However, the common Cys273 allele of rs2230624 is associated with increased risk, suggesting that loss of CD30 function may be protective for asthma. This hypothesis would be consistent with data showing that the Tyr273 leads to reduced cellular and soluble CD30 levels[13] and that CD30 increases pro-inflammatory cytokine production by CD4$^+$ T cells in the context of Th1, Th2, and Th17 immune responses[44]. Furthermore, soluble CD30 levels are elevated in patients with asthma or atopic dermatitis and correlate with disease severity[45,46]. When gene expression data were used to prioritize candidate causal genes, *CD200R1* and *AHI1* emerged as two other potential genes of interest related to immune function, given the highly significant *cis* eQTLs ($P < 1.0 \times 10^{-300}$) observed in blood with the lead variants on chromosomes 3q13.2 (rs7622814) and 6q23.3 (rs4526212), respectively. For example, CD200:CD200R1 signaling suppresses expression of proinflammatory molecules[47] and the induction of CD8$^+$ T cells[48]. By comparison, *AHI1* may be involved in JAK-STAT signaling[49] and downstream effects on Th1 and Th2 cells[50]. In addition, a recent GWAS identified a variant (rs9647635) at the chromosome 6q23.3 locus in near perfect LD with our peak SNP (rs4526212; $r^2 = 0.98$) that was associated with selective IgA deficiency[51], which is correlated with asthma[52].

Another important aspect of our study is the in vivo functional validation of *CD52* as potential candidate causal gene at the chromosome 1p36.11 locus, which may point to a role for the pathway involving this immune cell membrane protein in the development of asthma. For example, the mouse αCD52 antibody led to depletion of several populations of CD45$^+$ cells, in particular CD4$^+$ and CD8$^+$ T cells and CD19$^+$ B cells. These effects were observed in the spleen, which in mice is considered as accurately reflecting the pool of peripheral immune cells[53], and mimicked the immunosuppressive activity of the human monoclonal αCD52 antibody, alemtuzumab, on the same populations of circulating lymphocytes in patients[22,23]. However, the mouse αCD52 antibody also depleted leukocytes in the lung and BAL, with T cells and eosinophils being reduced to a greater extent than neutrophils. This pattern is consistent with data in humans showing that CD52 is expressed highest on T and B lymphocytes, to a lesser extent on eosinophils and monocytes, and at very low levels or absent on neutrophils, natural killer cells, and hematopoietic stem cells[21]. Moreover, by depleting leukocytes in lung and BAL, the mouse αCD52 antibody also had biological effects at the tissue level that, to our knowledge, have not been reported with alemtuzumab in humans. Of most importance and relevance to asthma, mice receiving the αCD52 antibody exhibited significantly decreased allergen-induced AHR, pulmonary inflammation, and airway epithelium thickness. In humans, alemtuzumab is primarily used to treat patients with relapsed forms of lymphocytic leukemia or multiple sclerosis[24–29]. However, generally favorable long-term outcomes have been reported in small numbers of patients with hypereosinophilic syndrome as well[54,55]. It is important to note that serious, albeit very rare, side effects have been associated with the immunosuppressive effects of targeting CD52[21], which has led to dose-reducing strategies[56]

and development of other αCD52 antibodies that may be more efficacious but less immunogenic[57]. Since the precise mechanisms through which alemtuzumab exerts its therapeutic effects are still not entirely known, a better understanding of CD52's biological functions will be needed in order to determine its role in the development of asthma, particularly with respect to other subtypes besides those induced by allergens.

While the present analyses expand the number of genetic determinants for asthma, our study should also be taken in the context of its limitations. First, our inclusion of subjects in the UK Biobank with self-reported asthma could have led to misclassification of cases and biased the results toward the null. However, the large sample size in our study and our ability to replicate all but three of the known 146 asthma loci indicate that our analyses were robust to the presence of such potential confounding factors. Second, recent studies have shown that both common and distinct loci underlie susceptibility to asthma in adulthood and in children[10,11], but we did not take age of onset into account for our analyses in order to increase power for detecting associations and due to the unavailability of this information in publicly available summary data from the TAGC. Third, since most subjects in the UK Biobank and TAGC were of European ancestry, some, but not all, of the genetic associations we identified may be relevant in non-European populations[9]. This notion has previously been observed with both common and rare variants even among closely related Latino populations and in comparison, to subjects of African ancestry[14,58]. Fourth, the lack of a replication dataset did not allow us to independently validate all of the 66 previously unknown loci for asthma. Fifth, the asthma loci identified in our meta-analysis relied on GWAS results in the TAGC that were derived from imputed Z scores and P-values. Although this approach allowed us to include the largest number of SNPs possible in the meta-analysis with the UK Biobank, it may not have provided the most precise summary statistics in the TAGC. The likelihood that this potential limitation affected our overall results was low since the Z-score meta-analysis replicated the association signals at nearly all known asthma susceptibility loci. Furthermore, even though the Z-score meta-analysis did not provide effect sizes, the ORs based on association tests in the UK Biobank alone are still likely to be very similar to those derived from the combined datasets given the large size of the UK Biobank. Finally, our study was primarily focused on discovery of main genetic effects of common susceptibility alleles. However, rare variants or GxE interactions, particularly with respect to other environmental exposures besides sex, may still play important roles in modulating asthma risk and will need to be addressed in appropriately designed future studies.

In summary, our large-scale genome analyses have expanded our understanding of the genetic basis of asthma and implicated sex and regulation of immune system processes as important components of disease susceptibility. These results provide opportunities for the evaluation of additional candidate causal genes and further exploration of the biological mechanisms underlying the pathogenesis of asthma.

## Methods

**Study populations**. Between 2006 and 2010, the UK Biobank recruited a total of 503,325 participants who were 40–69 years of age and registered with a general practitioner of the UK National Health Service (NHS)[59]. All study participants provided informed consent and the study was approved by the North West Multicentre Research Ethics Committee. Participating cohorts in the Trans-National Asthma Genetic Consortium (TAGC)[18] are summarized in Supplementary Data 1 and included 56 studies of European ancestry subjects (19,954 asthma cases, 107,715 controls), 7 studies of African ancestry subjects (2149 asthma cases, 6055 controls), two studies of Japanese subjects (1239 asthma cases, 3976 controls), and one study of Latino subjects (606 asthma cases, 792 controls), with a total of 23,948

asthma cases and 118,538 controls. There were 27 studies (8976 asthma cases, 18,399 controls) where cases were only defined based on having childhood-onset asthma (defined as asthma diagnosed at or before 16 years of age). All participants in the UK Biobank and TAGC gave written consent for participation in genetic studies, and the protocol of each study was approved by the corresponding local research ethics committee or institutional review board. The present study was approved by the Institutional Review Boards of the USC Keck School of Medicine.

**GWAS analyses in the UK Biobank.** All subjects in the UK Biobank were included in our analyses regardless of ancestry. Asthma cases were defined based on field code 6152_8 (doctor diagnosed asthma), International Classification of Diseases version-10 (ICD10) J45 (asthma)/J46 (severe asthma), and self-reported asthma. Field 6152 is a summary of the distinct main diagnosis codes a participant had recorded across all their hospital visits. Non-asthmatic controls were defined as individuals free from field code 6152_8 (doctor diagnosed asthma) as well as field code 6152_9 (doctor diagnosed allergic diseases), ICD10 J45/J46/J30 (hay fever)/L20 (dermatitis and eczema), and self-reported asthma/hay fever/eczema/allergy/allergy to house dust mite (HDM). This strategy resulted in a broad definition of asthma with 64,538 cases and 329,321 controls in the UK Biobank, and was selected to increase the likelihood of identifying asthma-associated loci. Quality control of samples, DNA variants, and imputation were performed by the Wellcome Trust Centre for Human Genetics[59]. Briefly, ~90 million SNPs imputed from the Haplotype Reference Consortium, UK10K, and 1000 Genomes imputation were available in the UK Biobank. Of these, 9,572,556 variants were used for GWAS analysis after filtering on autosomal SNPs with INFO scores >0.8 (directly from the UK Biobank) and with minor allele frequencies (MAF) >1% in the 487,409 individuals with imputed genotypes. A GWAS analysis was performed with BOLT-LMM v2.3.2 using a standard (infinitesimal) mixed model to correct for structure due to relatedness, ancestral heterogeneity, with adjustment for age, sex, the first 20 principal components, and genotyping array[60]. The genome-wide significance threshold was set at $P = 5.0 \times 10^{-8}$. Quantitative SNP effect size estimates obtained from BOLT-LMM were transformed to odds ratios (ORs) and standard errors (SEs) using the following formula: β or SE/(μ * (1 − μ)), where μ = case fraction[60]. In addition to the genomic control factor (λ), we also used LD Score regression to evaluate stratification in the UK Biobank GWAS results. This approach can provide a more accurate correction factor than genomic control in GWAS with large sample sizes and help distinguish between inflation that is due to true genetic signals from inflation that is due to stratification[61]. Although LD Score regression assumes a homogenous population, it could still be applicable to the UK Biobank since most subjects are of European ancestry. Manhattan and quantile-quantile plots were constructed using 'qqman' R package (v0.1.4)[62].

**Imputation in the TAGC.** Publicly available GWAS summary statistics for asthma with 2,001,281 SNPs in multi-ancestry populations and in only subjects of European-ancestry from the TAGC[18] were imputed with summary statistics imputation (SSimp) (v0.5.6) software[63] using the European population from the 1000 Genomes Project (Phase 1 release, v3) as a reference panel for LD computation. This imputation resulted in Z-scores and P-values for association of 16,918,874 SNPs with asthma in multi-ancestry and European-ancestry populations from the TAGC. After filtering on autosomal SNPs with MAF >1%, 9,415,011 variants were available for a meta-analysis with the UK Biobank.

**Meta-analyses for asthma in the UK Biobank and TAGC.** We performed a Z-score meta-analysis for asthma by combining the imputed Z-score and P-value summary level data in the TAGC with the results of our linear mixed model GWAS for asthma in the UK Biobank, as implemented in METAL[64]. The meta-analysis included 8,365,359 SNPs common to both datasets and assumed an additive model. Similar to the GWAS analysis in the UK Biobank alone, the genomic control factor (λ) and LD Score intercept were used to evaluate stratification in the meta-analysis results. We also performed a fixed-effect meta-analysis for asthma with betas and standard errors obtained in our GWAS analysis of the UK Biobank and those provided in the TAGC for 1,978,494 SNPs common to both datasets. The genome-wide threshold for significant association in both meta-analyses was set at $P = 5.0 \times 10^{-8}$ and replication of known asthma/allergic diseases loci in our meta-analysis was considered significant at a Bonferroni-corrected threshold of $P = 3.4 \times 10^{-4}$ for testing 146 loci (0.05/146). A locus was defined as not previously known if our sentinel SNP was >1 Mb away or in weak or no linkage disequilibrium (LD; $r^2 \le 0.1$) with the lead variants at the 146 previously reported loci for asthma and/or allergic diseases. To investigate whether the asthma loci identified in our meta-analysis harbored multiple distinct signals, we used FUMA (v1.3.5)[65], which is an integrative post-GWAS annotation web-based tool that defines independent variants using LD information from the 1000 Genomes Project. Based on the most significant SNPs at each of the 66 loci, secondary independent lead SNPs were defined as those yielding $P < 5.0 \times 10^{-8}$, in low LD ($r^2 < 0.1$) with the primary variant, and closely located to each other (<1 Mb based on the most right and left SNPs from each LD block). Manhattan and quantile-quantile plots were constructed using 'qqman' R package (v0.1.4)[62].

**Heritability.** Heritability was calculated with the INDI-V calculator under multifactorial liability threshold model. The baseline population risk (K) was set at 15%[5] and twin heritability ($h_L^2$) was set at 65% (average of 35 and 95%)[3]. GWAS results from the UK Biobank were used to estimate the heritability attributable to asthma-associated variants at the 66 loci.

**Sex-stratified associations with asthma susceptibility loci.** Sex-stratified association with variants at all 212 loci and risk of asthma was carried out with primary level data in the UK Biobank using logistic regression in males and females separately, adjusted for age, the first 20 principal components, and genotyping array. Logistic regression with the inclusion of a SNP * sex interaction term was also used to test for an interaction between SNP and sex on risk of asthma according to the following model: Logit [p] = $\beta_0 + \beta_1$ SEX + $\beta_2$ SNP + $\beta_3$ SEX * SNP. Interaction P-values (P-int) were considered significant at the Bonferroni-corrected threshold for testing 212 loci (0.05/212 = $2.4 \times 10^{-4}$). Sex-stratified and gene-sex interaction analyses were performed with STATA (v15.0, StataCorp LP, Texas, USA).

**Genetic risk score analyses.** Primary level data from the UK Biobank were used to generate weighted genetic risk scores (GRS) for the 66 previously unreported, 146 known, and all 212 loci. For each variant, the number of risk alleles was multiplied by its respective weight (i.e., the natural log of the OR) obtained from the GWAS analysis in the UK Biobank and summed together across all variants to generate the three GRS. Sex-specific weighted GRS were also constructed for the 66 previously unreported, 146 known, and all 212 loci using effect sizes obtained from the logistic regressions carried out in males and females separately. Association between weighted GRS and asthma in all subjects and in males and females separately was tested by grouping participants into deciles according to the distribution of the weighted GRS and using logistic regression to calculate ORs for subjects in the highest decile compared to the middle and lowest deciles, with adjustment for age, sex (if applicable), the first 20 principal components, and genotyping array. Tests for an interaction between sex and the GRS on risk of asthma were carried out by including a sex * sex-specific GRS interaction term in the logistic regression comparing the highest versus lowest decile or by including the sex-interaction term in a model across all deciles, with inclusion of the same covariates. GRS-based sex-stratified and gene-sex interaction analyses were performed with STATA (v15.0, StataCorp LP, Texas, USA).

**Sensitivity analyses.** Sensitivity analyses were carried out in the UK Biobank to compare the association signals at the 66 loci for the broad and all-inclusive definition of asthma (described above) with two more strict definitions of asthma, as well as with allergic diseases only. The first strict definition of asthma included cases without allergic diseases ($n = 33,830$) based on subjects being positive for only field code 6152_8 (doctor diagnosed asthma) or ICD10 J45 (asthma)/J46 (severe asthma) or self-reported asthma (20002_1111), and negative for field code 6152_9 (doctor diagnosed allergic diseases), ICD10 J30 (hay fever) /L20 (dermatitis and eczema) and self-reported hay fever/eczema/allergy/allergy to house dust mite. The second strict definition of asthma defined cases ($n = 56,820$) by only including subjects positive for field code 6152_8 (doctor diagnosed asthma) or ICD10 J45 (asthma)/J46 (severe asthma) or self-reported asthma (20002_1111) and negative for chronic obstructive pulmonary disease (COPD), emphysema, or chronic bronchitis (doctor diagnosed 22128, 22129 and 22130, self-reports 20002_1112 and 20002_1113 or ICD-10 codes J43, J430, J431, J432, J438, J439, J44, J440, J441, J448, and J449). Allergic disease-specific cases ($n = 93,468$) were defined as positive for field code 6152_9 (doctor diagnosed allergic diseases), ICD10 J30 (hay fever)/L20 (dermatitis and eczema), or self-reported hay fever (20002_1387) /eczema (20002_1452)/allergy (20002_1374)/allergy to house dust mite (20002_1668), and negative for field code 6152_8 (doctor diagnosed asthma), ICD10 J45/J46, and self-reported asthma (20002_1111). Controls ($n = 329,321$) were defined as individuals free from field codes 6152_8 (doctor diagnosed asthma), 6152_9 (doctor diagnosed allergic diseases), ICD10 J45/J46/J30/L20, and self-reported hay fever (20002_1387)/eczema (20002_1452) /allergy (20002_1374) /allergy to HDM (20002_1668). Lastly, we also carried out a GWAS for broadly defined asthma in the UK Biobank with only subjects of European ancestry ($n = 53,924$ cases and 276,523 controls) to assess whether there was any potential confounding due to population structure.

**Association with lung function and other disease traits.** Association of asthma loci with measures of lung function was evaluated in 443,466 subjects from the UK Biobank. Spirometry traits used for these analyses were obtained at the initial assessment visit at which participants were recruited between 2006 and 2010 (age range 37–73; mean ± SD = 56 ± 8) and included forced expiratory volume in 1-second (FEV$_1$; field codes 3063), forced vital capacity (FVC, field code 3062), FEV$_1$/FVC, and peak expiratory flow (PEF, field code 3064). Analyses were performed with BOLT-LMM (v2.3.2) using a standard (infinitesimal) mixed model to correct for structure due to relatedness, ancestral heterogeneity, or other factors, with adjustment for age, sex, the first 20 principal components, genotyping array, height and smoking status (ever vs. never). All lung function traits were inverse-normal transformed prior to analysis and the threshold for the significant association was

based on a Bonferroni correction for testing 66 loci ($P = 0.05/66 = 7.6 \times 10^{-4}$). Evaluation of the asthma loci for association with other disease traits (PheWAS) was carried out using the Phenoscanner database[19]. Only associations with $P < 5.0 \times 10^{-8}$ and derived from SNPs in high LD ($r^2 > 0.8$) with our lead GWAS variants are reported for the PheWAS analyses.

**Enrichment of asthma loci in epigenetic marks**. Enrichment of asthma-associated variants in DNase I hypersensitive sites (peaks) was determined using the GARFIELD (v2) method[66]. GARFIELD leverages GWAS findings with regulatory or functional annotations (primarily from ENCODE and Epigenomics Roadmap data)[67] to find features relevant to a phenotype of interest. Briefly, the method first uses a greedy procedure to extract a set of independent variants from the genome-wide genetic variants, using LD ($r^2 \geq 0.01$) and distance information (LD pruning step). Second, it annotates each variant with a regulatory annotation if either the variant, or a correlated variant ($r^2 \geq 0.8$), overlaps the feature (LD tagging annotation step). Third, it calculates ORs and enrichment p-values at different GWAS p-value thresholds for each annotation using a logistic regression model with 'feature matching' on variants by distance to the nearest TSS and number of LD proxies ($r^2 \geq 0.8$). In addition to GARFIELD, we also performed gene prioritization and tissue enrichment analyses using DEPICT (v1.1)[68] with 33,017 genome-wide significant SNPs ($P < 5.0 \times 10^{-8}$) associated with asthma that were identified in the multi-ancestry meta-analysis with the UK Biobank and TAGC. Both nominal P-values and false discovery rates (FDRs) were calculated for gene set enrichment and tissue enrichment, as implemented in DEPICT[68].

**Expression quantitative trait locus (eQTL) analyses**. Functional evaluation of SNPs at the 66 previously unreported asthma loci and prioritization of candidate causal genes was determined using multi-tissue eQTL data from the GTEx Project (version 6)[69], the eQTLGen Consortium (https://www.eqtlgen.org/), the PhenoScanner database[19], and various other publicly available sources (Supplementary Data 13). Consideration was only given to *cis* eQTLs with $P < 5.0 \times 10^{-8}$ that were derived from our lead GWAS variants or proxy SNPs in high LD ($r^2 > 0.8$).

**Animal husbandry**. Female BALB/cByJ mice between 6 and 8 weeks of age were purchased from Jackson Laboratory (Bar Harbor, Maine) and housed 4–5 per cage at 25 °C on a 12 h dark/12 h light cycle with ad libitum access to food (Purina chow diet #5053) and water. All animal studies were approved by the USC Keck School of Medicine Institutional Animal Care and Use Committee and conducted in accordance with the Department of Animal Resources' guidelines.

**Functional evaluation of CD52**. Mice were immunized on day 1 with 100 µg of house dust mite (HDM) in 2 mg of aluminum hydroxide (alum) by intraperitoneal (i.p.) injection. On day 7, mice were randomly assigned to three groups. Group 1 was intraperitoneally administered 500 µg of a monoclonal αCD52 IgG2a depleting antibody (clone BTG-2G, Cat. # D204-3L2, Lot # 004; MBL International, Woburn, MA) and Group 2 was intraperitoneally administered 500 µg of an IgG2a isotype control antibody (clone 2A3, Cat. # BE0085, Lot # 63955A2; BioXCell, West Lebanon, NH). Group 3 only received PBS as a control. On days 8, 9, and 10, mice in Groups 1 and 2 were intravenously (i.v.) administered 250 µg of the αCD52 antibody or the isotype control antibody, respectively, and simultaneously challenged intranasally (i.n.) with 50 µg HDM. Mice in Group 3 were only challenged intranasally with PBS on days 8, 9, and 10. On day 11, mice underwent invasive plethysmography to measure AHR. Lung resistance and dynamic compliance at baseline and in response to sequentially increasing doses of methacholine (5–40 mg/ml) was measured using a Buxco FinePointe (Data Sciences International, St. Paul, MN) respiratory system in tracheostomized immobilized mice that were mechanically ventilated under general anesthesia[17]. AHR values were recorded during a 3-min period during each methacholine dose challenge. After AHR measurements were completed, mice were euthanized for collection of tissues and flow cytometric analysis of immune cells. The trachea was cannulated and 2 ml of PBS was used to collect bronchial alveolar lavage (BAL) fluid, followed by removal of the lungs and spleen.

**Flow cytometry**. Single cell suspensions were prepared from lung and spleen tissue in accordance with standard protocols. Immune cells were quantified by means of flow cytometry after staining cells with a 1:200 dilution of the following antibodies: allophycocyanin (APC)/Cy7-anti-CD45 (clone 30-F11, Cat. # 103116, Lot # B287097; BioLegend, San Diego, CA), fluorescein isothiocyanate (FITC)-anti-CD19 (clone MB19-1, Cat. # 101506, Lot # B215595; BioLegend, San Diego, CA), peridinin-chlorophyll-protein complex (PerCP)/Cy5.5-anti-CD3ε (clone 17A2, Cat. # 100218, Lot # 260625; BioLegend, San Diego, CA), brilliant violet 421™ (BV)-anti-CD4 (clone GK1.5, Cat. # 100438, Lot # B159950; BioLegend, San Diego, CA), PE/Cy7-anti-CD8 (clone 53-6.7, Cat. # 25-0081-82, Lot # 4290706; eBioscience, San Diego, CA) in the presence of anti-mouse Fc-block (clone 2.4G2, Cat. # BE0307; BioXcell, West Lebanon, NH). Leukocytes in BAL fluid were quantified after staining cells with PE-anti-Siglec-F (clone E50-2440, Cat. # 552126, Lot #8081855, BD Biosciences, San Jose, CA), PE/Cy7-anti-CD45 (clone 30-F11, Cat. # 103114, Lot # B219150; BioLegend, San Diego, CA), APC/Cy7-anti-CD11c (clone N418,

Cat. # 117324, Lot B263881; BioLegend, San Diego, CA), FITC-anti-CD19 (clone MB19-1, Cat. # 101506, Lot # B215595; BioLegend, San Diego, CA), PerCP/Cy5.5-anti-CD3ε (clone 17A2, Cat. # 100218, Lot # 260625; BioLegend, San Diego, CA), APC-anti-Gr-1 (clone RB6-8C5, Cat. # 108412; Lot # B277576; BioLegend, San Diego, CA), and eFluor450-anti-CD11b (clone M1/70, Cat. # 48-0112-82, Lot # 4329941; eBioscience, San Diego, CA) in the presence of anti-mouse Fc-block (clone 2.4G2, Cat. # BE0307, Lot # 827594A9; BioXcell, West Lebanon, NH). The gating strategy used with these antibodies is shown in Supplementary Fig. 4. CountBright absolute count beads (Thermo Fisher Scientific, Waltham, Mass) were used to calculate absolute cell number, according to the manufacturer's instructions. At least $10^5$ CD45$^{++}$ cells were acquired on a BD FACSCanto II (BD Biosciences, San Jose, CA). Data were analyzed with FlowJo (v10) software (TreeStar, Ashland, OR).

**Histology**. Lungs were harvested, fixed overnight in 4% paraformaldehyde, and embedded in paraffin. 4 µm sections were cut and stained with H&E according to standard protocols. Images of the H&E-stained slides were acquired with a KeyenceBZ-9000 microscope (Keyence, Itasca, Il) and analyzed for the number of inflammatory cells and thickness of the airway epithelium with the ImageJ Analysis Application (v2) (NIH & LOCI, University of Wisconsin).

**Statistical analysis**. Differences in measured variables between groups of mice were determined by unpaired Student's *t*-test (SAS 9.3; SAS Institute Inc) and considered statistically significant at $P < 0.05$. Data are expressed as mean ± SE.

## Data availability
Summary statistics for the GWAS analysis in the UK Biobank and the meta-analysis with TAGC are available at The NHGRI-EBI Catalog of published genome-wide association studies: [https://www.ebi.ac.uk/gwas/]. All other data supporting the findings of this study are available either within the article, the Supplementary Information and Supplementary Data files, or upon reasonable request. The Source Data file for Figs. 3–5 and Supplementary Fig. 5 is available at: [https://doi.org/10.6084/m9.figshare.11955108]. Publicly available resources used for this study were The UK Biobank [http://www.ukbiobank.ac.uk/]; Trans-National Asthma Genetic Consortium [https://www.ebi.ac.uk/gwas/downloads/summary-statistics]; Genotype-Tissue Expression Portal [http://gtexportal.org/]; eQTLGen Consortium [https://www.eqtlgen.org/]; and the Phenoscanner database [http://www.phenoscanner.medschl.cam.ac.uk/phenoscanner].

## Code availability
All analyses described in this study used publicly available software: Summary statistics imputation (SSimp) [https://github.com/zkutalik/ssimp_software]; METAL [http://csg.sph.umich.edu/abecasis/Metal/]; FUMA [https://fuma.ctglab.nl/]; GARFIELD [https://www.ebi.ac.uk/birney-srv/GARFIELD/]; DEPICT [https://data.broadinstitute.org/mpg/depict/index.html]; R statistical software [http://www.R-project.org/].

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

## Acknowledgements

This work was supported, in part, by National Institutes of Health Grants R01HL133169, R01ES021801, R01ES025786, R01ES022282, R01HL144790, R21ES024707, R21AI109059, P30ES007048, and P01ES022845, and U.S. EPA Grant RD83544101. Use of the UK Biobank Resource was carried out under Application Number 33307. The funders had no role in study design, data collection and analysis, decision to publish, or preparation of the paper.

## Author contributions

Concept and design: Y.H., J.A.H., and H.A. Acquisition, analysis, and interpretation of data: Y.H., Q.J., P.S.J., B.P.H., C.P., P.H., J.G., N.C.W., E.E., F.D.G., O.A., J.A.H., and H.A. Drafting of the paper: Y.H., J.A.H., and H.A. Critical revision of the paper for important intellectual content: all authors.

## Competing interests

The authors declare no competing interests.
