## [Peer Review File · Nature Communications]

Reviewers' Comments:

Reviewer #1:

Remarks to the Author:

The authors have done a really good job at summarizing the analyses performed and the results obtained. Methods are very clear and sound, and I congratulate the authors for the amount of work done and the implications that might follow-up.

Functional analyses and in vivo studies are outside this reviewer's comfort zone. I have based my review on my knowledge of statistical methods to analyse asthma and lung function traits in the context of genetic epidemiology.

I have a couple of general points:

A basic description of UKBB and TAGC participants is missing; For instance, age of UKBB participants was 40-69 years, but what was the age range in TAGC participants? TAGC study includes 27 studies with only childhood-onset asthma (defined as asthma diagnosed at or before 16 years of age). It would be good if the authors could mention the role of age of asthma onset in the genetic architecture of asthma and how this is a limitation in this study.

The authors mention two reasons for the still small fraction of the overall heritability of asthma that is explained by risk alleles discovered so far: undiscovered rare variants and/or gene-by-environment interactions. Could the use of a qualitative trait which does not depict the spectrum of asthma heterogeneity be another reason?

Some other minor comments follow

Results

Lines 94-99. Supp. Figure 1 reports 167 loci instead of 145 as reported in text and Supp. Table 2 "The remaining 104 overlapped with the 145 loci previously identified.." this last sentence is confusing since you just said that 41 of 145 are novel

How many of these loci are independent signals? TAGC study identified 22.

Methods

Line 405. How was the SE for the OR calculated?

Lines 470-475. At what age were lung function measures taken? Why were FEV1 and FVC log-transformed and were analyses with FEV1 and FVC adjusted for age, sex and height?

Typos

Line 269 "...progress in this area been limited"

Line 308 "...downstream effects on Th1 and Th2 cells."

What does PBS stands for?

References 21 and 30 are duplicates

Reviewed by R Granell

Reviewer #2:

Remarks to the Author:

The authors should be commended in their use of publicly available data to identify additional novel risk variants for asthma. In addition to some minor comments, I have some major concerns that would need to be addressed before this manuscript is suitable for publication.

Major concerns:

- Section on "Sex-stratified and Polygenic Risk Score Analyses with Asthma Susceptibility Loci:"
The genetic risk scores are based on the 100s of variants discovered in this analysis, and not considering all SNPs (e.g. using LDpred), and should therefore be referred to throughout the text as a genetic risk score, and NOT a "polygenic" risk score. The authors compare the risk of disease between the highest and lowest decile, however, it is common practice to compare the risk between the highest and middle decile. Furthermore, the loci was identified using the same UKBiobank samples that the genetic risk score was then calculated for, which most likely will result in inflated conclusions. A risk score calculated from the lead variants identified in the TAGC GWAS, or a polygenic risk score constructed from the TAGC summary statistics, would be more appropriate. Alternatively, the authors could exclude all results comparing risk between subjects, and only retain the genetic risk score and interaction results, if the intent is to show that at an aggregate level for the variants identified in their analysis, risk may be different between the sexes.
- As the authors did not provide evidence for replication of the 66 novel loci, and they analyzed ~10 million SNPs which approaches GWAS of WGS data, I suggest that at the very least, the authors report how many of the loci are novel at a threshold of 5×10^{-9} , as recently suggested [PMID: 30623491]. The authors should also list lack of a replication data set as a limitation of their study. (The authors should be commended for their functional validation of one of the novel results, but as this functional validation was performed for only one of the results, lack of replication is a limitation of this study.)
- Lines 408-412: It sounds like the TAGC multi-ethnic meta-analysis results were used for imputation against an European LD reference panel. I suggest a sensitivity analysis where only the TAGC European summary statistics results are imputed and meta-analyzed with the UK Biobank. (Since ~90% of the TAGC subjects are European ancestry, I anticipate that results would mostly be the same, but it would be good to show this since technically the results could be inaccurate due to the LD reference panel only representing European ancestry subjects)
- The authors did not state whether they included all UK Biobank subjects in their analyses, or restricted their analysis to European ancestry subjects. The inflation factor of 1.31 reported with supplementary figure 2 suggests that this may be the case (this reviewer downloaded 20002_1111.ukbb.sumstats.gz from <http://ldsc.broadinstitute.org/gwashare/> and for that asthma GWAS, the inflation factor was 1.15). The authors should state explicitly whether they used European ancestry subjects only, and if not, potentially limit their analysis to European ancestry subjects or ensure that their results are not confounded by ancestry. If a large inflation factor is still observed, the authors should consider adjusting their results for the inflation factor. In addition, the authors managed to identify 64,538 asthma cases and 329,321 controls in the UK Biobank - the proportion of asthmatics seem very high, especially in comparison to the recent Pividori analysis (PMID: 31036433) where they identified 37,846 and 318,237 controls. At the very least, a sensitivity analysis of a more stringent case-control definition, such as the definition used by Pividori, should be performed.

Minor comments

- Line 31: "UK Biobank" should be "the UK Biobank" (and elsewhere in the text)
- Line 37: chromatic should be chromatin?
- Line 67-68: I would not make such a strong statement based on a single study (of relatively small sample size). Rather say something that only one study investigated rare variation, and that its results did not appear to indicate a large role for rare variation in asthma susceptibility.
- Lines 74-79: Adding a brief sentence on the main analyses done would be helpful to frame the content of the paper for the reader (the paragraph currently summarizes the data sets used and the main findings but not really what was done)
- Line 112: "region" - from my reading of Supplemental Table 4 you tested single variants and not regions?
- Lines 166-169: Asthma is an allergic disease (well, mostly). The terminology here is confusing: is this non-allergic asthma? Asthma with and without the presence of other allergic disease? This

seems to be explained later in the methods section but including a brief summary here would be useful to the reader.

- Figure 4 and Supplementary Table 8: enrichment for epithelium and thymus is not immediately evident to me from Figure 4 or Supplementary Table 8. In addition, Supplementary Table 8 contains 8,000 rows. I suggest submitting this instead as a supplementary data file.
- Lines 198-199: For reproducibility, it is important to state which public eQTL databases were used (e.g. list them in a supplementary table referred to from here).
- Lines 204-205: "blood as playing an important role in asthma" - please rephrase - it is probably immune cells in blood that play a role in asthma, not blood itself!
- Line 255: "in the" should be "the"
- Lines 265-267: references to support these statements?
- Line 345-346: " it is possible that the genetic association results may not be generalizable to other populations" - I would rephrase this as some but not all genetic association results may be relevant in non-European populations [PMID: 30787307]
- Figure 5, lines 513-519: How many mice were included in the experiment?
- Supplementary Figure 2: Please add the inflation factor at the very least to the figure legend text

Reviewer #3:

Remarks to the Author:

In this manuscript, Han et al. identify 66 novel genomic regions that are associated with asthma susceptibility, and colocalize to regions of open chromatin in immune cells. The authors focus on one candidate causal gene, CD52, which is expressed on a variety of immune cells and is the target of an FDA-approved human α -CD52 antibody, alemtuzumab. To test the in vivo effects of CD52 targeting, a CD52 antibody was administered during HDM exposure in mice, which reduced allergen-induced airway hyperreactivity, consistent with the authors' hypothesis that CD52 plays a role in inflammatory airway disease pathogenesis.

This study identifies a potential lead in asthma treatment, however, the in vivo mechanistic insight for how CD52 may be participating in immune priming or allergic airway disease is quite limited. For instance, which mouse/human immune cells express CD52 in the disease setting? Are they particular subsets that would have relevance to allergic disease? The authors mention antibody staining in the methods but this is not shown. In the HDM system, type 2 cytokines contribute to both innate and adaptive phases of the response but these are not evaluated. How are the human mutations that the authors connect genetically to disease contributing to function? Depletion of several immune cell types suggest that this effect is broadly immunosuppressive, consistent with prior human studies and the reduction in immune parameters.

Response to Reviewer 1's Comments

We would like to thank the Reviewer for taking the time to review our manuscript and providing helpful critique/comments. We have addressed all the points raised and hope these revisions satisfactorily answer his/her concerns. All changes to the manuscript are indicated by blue font and Page and Line numbers.

1. Comment: The authors have done a really good job at summarizing the analyses performed and the results obtained. Methods are very clear and sound, and I congratulate the authors for the amount of work done and the implications that might follow-up.

Response: We appreciate the positive feedback.

2. Comment: A basic description of UKBB and TAGC participants is missing; For instance, age of UKBB participants was 40-69 years, but what was the age range in TAGC participants? TAGC study includes 27 studies with only childhood-onset asthma (defined as asthma diagnosed at or before 16 years of age).

Response: We have now included a basic description of the UK Biobank and TAGC in the Methods section (Page 19, Lines 404-416). We have also provided a new Supplemental Table 2 describing the cohorts that were part of the TAGC, including age ranges.

3. Comment: It would be good if the authors could mention the role of age of asthma onset in the genetic architecture of asthma and how this is a limitation in this study.

Response: The Reviewer raises an excellent point. The focus of our study was to identify loci for asthma, defined broadly, although it is possible that genetic susceptibility to asthma in adulthood differs from that in children. This notion is supported by two recent studies that carried out separate GWAS analyses for childhood and adult-onset asthma in the UK Biobank, which identified both common and distinct loci for each asthma subtype (PMIDs: 30929738, 31036433). Thus, we do acknowledge that our study is limited in this regard since we did not take age of onset into account in our GWAS meta-analysis. This point has now been added to the limitations paragraph in the Discussion section (Page 17, Lines 373-376).

4. Comment: The authors mention two reasons for the still small fraction of the overall heritability of asthma that is explained by risk alleles discovered so far: undiscovered rare variants and/or gene-by-environment interactions. Could the use of a qualitative trait which does not depict the spectrum of asthma heterogeneity be another reason?

Response: We agree with the Reviewer that the use of a qualitative outcome, which may not capture the heterogeneous nature of asthma, could be another reason for the relatively small fraction of the heritability explained by known genetic determinants. We have added this point

to the Introduction section (Page 3, Lines 70-71).

5. Comment: Lines 94-99. Supp. Figure 1 reports 167 loci instead of 145 as reported in text and Supp. Table 2. “The remaining 104 overlapped with the 145 loci previously identified...” this last sentence is confusing since you just said that 41 of 145 are novel.

Response: We apologize for the confusion. The data referred to by the Reviewer in Supplemental Figure 1 and Supplemental Table 3 (formerly Supplemental Table 2) are the GWAS results for asthma in the UK Biobank alone. This analysis did identify 145 significantly associated loci, of which 41 were novel. The remaining 104 loci identified in the UK Biobank overlapped with 145 previously known loci (the fact that “145” came up twice is simply coincidence). The subsequent meta-analysis combining the GWAS summary statistics from the UK Biobank with those from the TAGC resulted in the identification of 167 loci for asthma, which are presented in Figure 2 and Supplemental Table 4. We hope this clarifies the Reviewer’s concern.

6. Comment: How many of these loci are independent signals? TAGC study identified 22.

Response: To identify independent association signals at the 66 novel asthma loci in addition to the lead variants, we used the FUMA program, which is an integrative post-GWAS annotation web-based tool that defines independent lead SNPs at loci of interest using linkage disequilibrium information from the 1000 Genomes Project (PMID: 29184056). Based on the most significant SNPs at each novel locus, FUMA defines secondary SNPs if they are independent from each other at an $r^2 < 0.1$ with $P < 5.0 \times 10^{-8}$, and closely located to each other (<1Mb based on the most proximal and distal SNPs in each LD block). Our analyses with FUMA revealed 12 secondary independent SNPs at 10 of the novel asthma loci. These results are described in the Results section (Page 7, Line 136-137) and in a newly provided Supplemental Table 7. The methodology for these analyses is also provided in the Methods section (Page 22, Lines 471-477).

7. Comment: Line 405. How was the SE for the OR calculated?

Response: SNP effect size estimates for asthma obtained from the BOLT-LMM analyses were transformed to ORs and SEs using the following formula: β or $SE/(\mu * (1-\mu))$, where μ =case fraction. This information has now been added to the Methods section (Page 20, Lines 441-443).

8. Comment: Lines 470-475. At what age were lung function measures taken? Why were FEV₁ and FVC log-transformed and were analyses with FEV₁ and FVC adjusted for age, sex and height?

Response: Subjects in the UK Biobank underwent lung function measurements at the initial assessment and recruitment visit, which took place between 2006-2010. Therefore, the ages at

which FEV₁, FVC, FEV₁/FVC and PEF (peak expiratory flow) were measured ranged from 37-73, with a mean \pm SD of 56 ± 8 . Furthermore, our initial analyses with FEV₁ and FVC only included age and sex as covariates in the analyses and used log-transformed values since these lung function traits did not exhibit normal distributions. However, in response to the Reviewer's comment, we re-evaluated the transformed lung function data, which still revealed slightly non-normal distributions. Therefore, we used inverse-normal transformed data for FEV₁ and FVC and re-analyzed the data with the inclusion of height as well as smoking status as two additional covariates. We also considered the ratio of FEV₁/FVC and peak expiratory flow (PEF) as two additional lung function traits, which we analyzed in the same manner as FEV₁ and FVC. All of this information is now included in the Methods (Page 25, Lines 541-551) and the updated association results with the 66 novel loci and all four lung function traits (FEV₁, FVC, FEV₁/FVC and PEF) are presented in the Results (Page 9, Lines 195-199) and a revised Supplemental Table 10.

9. Comment: Line 269 "...progress in this area been limited"
Line 308 "...downstream effects on Th1 and Th2 cells."

Response: These typographical errors have been fixed (Page 13, Line 291, and Page 15, Line 331).

10. Comment: What does PBS stands for?

Response: PBS stands for "phosphate-buffered saline" and has been clarified in the Results (Page 12, Line 261) and Legends to Figure 5 and Supplemental Figures 4 and 5.

11. Comment: References 21 and 30 are duplicates.

Response: Thank you for bringing this to our attention; reference 30 has been removed.

Response to Reviewer 2's Comments

We would like to thank the Reviewer for taking the time to review our manuscript and providing helpful critique/comments. We have addressed all the points raised and hope these revisions satisfactorily answer his/her concerns. All changes to the manuscript are indicated by blue font and Page and Line numbers.

1. Comment: The authors should be commended in their use of publicly available data to identify additional novel risk variants for asthma.

Response: We appreciate the positive feedback.

2. Comment: The genetic risk scores are based on the 100s of variants discovered in this analysis, and not considering all SNPs (e.g. using LDpred), and should therefore be referred to throughout the text as a genetic risk score, and NOT a "polygenic" risk score.

Response: As requested, we now refer to cumulative genetic burden throughout the text as genetic risk score (GRS) rather than polygenic risk score.

3. Comment: The authors compare the risk of disease between the highest and lowest decile, however, it is common practice to compare the risk between the highest and middle decile.

Response: We carried out the analysis requested, which demonstrated that individuals in the highest decile for risk alleles at all 211 loci still had a highly significant 2.5-fold increased risk of asthma compared to subjects in the middle decile (OR=2.5, 95% CI 2.4-2.6; $P < 1.0 \times 10^{-305}$). This information has now been added to the Results section (Page 8, Lines 159-160). A significant interaction with sex was also observed ($P_{\text{int}} = 3.5 \times 10^{-6}$), where the increased risk of asthma for men in the highest vs. middle decile (OR=2.7, 95% CI 2.6-2.9; $P = 5.1 \times 10^{-295}$) was still to an even greater extent than that for women in the highest vs. middle decile (OR=2.3, 95% CI 2.2-2.4; $P = 1.4 \times 10^{-273}$).

4. Comment: Furthermore, the loci were identified using the same UK Biobank samples that the genetic risk score was then calculated for, which most likely will result in inflated conclusions. A risk score calculated from the lead variants identified in the TAGC GWAS, or a polygenic risk score constructed from the TAGC summary statistics, would be more appropriate. Alternatively, the authors could exclude all results comparing risk between subjects, and only retain the genetic risk score and interaction results, if the intent is to show that at an aggregate level for the variants identified in their analysis, risk may be different between the sexes.

Response: The primary goal of the sex-stratified follow up analyses was to determine whether risk of asthma was different in men and women with respect to the 211 known and newly identified loci, either individually or in aggregate. We elected to use the UK Biobank for this

purpose since we had access to primary level sex, genotype, and phenotype data. These analyses revealed four previously known loci on chromosomes 1p34.3, 2q37.3, 5q22.1, and 6p25.3 that individually yielded evidence for Bonferroni-corrected significant gene-sex interactions. As suggested by the Reviewer, we also evaluated whether cumulative genetic burden differed between the sexes. For example, we used sex-specific effect sizes to construct weighted genetic risk scores (GRS) in men and women separately for the 66 novel, 145 known, and all 211 asthma loci. This analysis revealed a pattern where risk of asthma among subjects in the highest versus lowest decile of the GRS was increased by approximately one order of magnitude greater in men than in women. These observations were also accompanied by significant gene-sex interactions when comparing the highest versus lowest decile for the GRS or the entire spectrum of cumulative genetic burden for the 66 novel, 145 known, and all 211 asthma loci. All of these data are described in the Results section (Page 7, Lines 147-150 and Page 8, 164-179) and presented in Figure 3B-D and Supplemental Table 8.

5. Comment: As the authors did not provide evidence for replication of the 66 novel loci, and they analyzed ~10 million SNPs which approaches GWAS of WGS data, I suggest that at the very least, the authors report how many of the loci are novel at a threshold of 5×10^{-9} , as recently suggested [PMID: 30623491]. The authors should also list lack of a replication data set as a limitation of their study. (The authors should be commended for their functional validation of one of the novel results, but as this functional validation was performed for only one of the results, lack of replication is a limitation of this study.)

Response: As requested, we now indicate that 32 of the 66 novel loci were associated with asthma at the more stringent significance threshold of $P=5.0 \times 10^{-9}$ (Page 6, Lines 120-122). We also include the lack of a replication dataset in the limitations paragraph in the Discussion section (Page 17, Lines 381-382).

6. Comment: Lines 408-412: It sounds like the TAGC multi-ethnic meta-analysis results were used for imputation against an European LD reference panel. I suggest a sensitivity analysis where only the TAGC European summary statistics results are imputed and meta-analyzed with the UK Biobank. (Since ~90% of the TAGC subjects are European ancestry, I anticipate that results would mostly be the same, but it would be good to show this since technically the results could be inaccurate due to the LD reference panel only representing European ancestry subjects)

Response: We thank the Reviewer for this excellent suggestion. As requested, we used the TAGC European summary statistics results for imputation using the European LD reference panel, which was then meta-analyzed with the UK Biobank. This European-specific meta-analysis yielded 32,697 genome-wide significant SNPs across 152 loci. Of these, 51 loci were novel and the remaining 101 were previously known. Of the 51 novel loci, 48 overlapped with the 58 novel loci identified in the meta-analysis we carried out using the multi-ethnic summary statistics in TAGC with all subjects from the UK Biobank. Thus, the meta-analysis suggested by the Reviewer identified nearly all the same novel loci. The other three loci were highly suggestively associated with asthma in the multi-ethnic meta-analysis but only achieved genome-wide significance in the meta-analysis that included subjects of European ancestry from the

TAGC. Conversely, the 10 other loci that were genome-wide significant in either the UK Biobank alone or in the multi-ethnic meta-analysis with TAGC and the UK Biobank still yielded suggestive evidence for significance in the European-only meta-analysis. This information is described in the Results section (Page 6, Lines 125-135), in Supplemental Table 6, and in the Methods section (Page 21, Lines 449-455).

7. Comment: The authors did not state whether they included all UK Biobank subjects in their analyses or restricted their analysis to European ancestry subjects. The inflation factor of 1.31 reported with supplementary figure 2 suggests that this may be the case (this reviewer downloaded 20002_1111.ukbb.sumstats.gz from <http://ldsc.broadinstitute.org/gwashare/> and for that asthma GWAS, the inflation factor was 1.15). The authors should state explicitly whether they used European ancestry subjects only, and if not, potentially limit their analysis to European ancestry subjects or ensure that their results are not confounded by ancestry. If a large inflation factor is still observed, the authors should consider adjusting their results for the inflation factor.

Response: We apologize for not being more specific. The summary statistics referred to by the Reviewer were from a GWAS for asthma carried out by the Neale lab in the UK Biobank using only subjects of white British genetic ancestry and excluding closely related individuals. This resulted in the inclusion of ~337,000 subjects in their analyses. By comparison, our sample size in the UK Biobank was significantly larger since we considered all subjects with imputed genotype data regardless of ancestry or relatedness (n=487,409) prior to defining cases and controls. As a result of this inclusion strategy, we used a linear mixed model (as implemented in BOLT-LMM) for the GWAS analysis to control for population structure due to relatedness, ancestral heterogeneity, and/or other factors. Despite using linear mixed models, such confounders can still have inflated the distribution of test statistics and resulted in the genomic control factor of 1.31 that we obtained from the GWAS results in the UK Biobank. Therefore, we also used LD Score regression as another approach to evaluate stratification. LD Score regression can provide a more accurate correction factor than genomic control in GWAS with large sample sizes and help distinguish between inflation that is due to polygenicity from inflation that is due to stratification (PMID: 25642630). The LD Score intercept from the BOLT-LMM GWAS analysis in the UK Biobank was not significantly greater than 1 (1.076; SE=0.009), suggesting that any inflation of test statistics was more likely due to many small genetic effects rather than population structure. This information provided in the Methods (Page 20, Lines 443-447) and the Legend to Supplementary Figure 1, which shows the GWAS results in the UK Biobank. Lastly, we also carried out the sensitivity analysis suggested by the Reviewer by testing the 66 novel loci for association with asthma in the UK Biobank with subjects of non-European ancestry excluded. As shown in a revised Supplemental Table 9, the association results in only subjects of European ancestry were comparable to those that included all subjects from the UK Biobank.

8. Comment: In addition, the authors managed to identify 64,538 asthma cases and 329,321 controls in the UK Biobank - the proportion of asthmatics seem very high, especially in comparison to the recent Pividori analysis (PMID: 31036433) where they identified 37,846 and

318,237 controls. At the very least, a sensitivity analysis of a more stringent case-control definition, such as the definition used by Pividori, should be performed.

Response: We thank the Reviewer for this suggestion. As noted above in our response to Point #7, we included all subjects in the UK Biobank with imputed genotype data prior to defining asthma cases and controls. Furthermore, to increase power for detecting associations, our primary GWAS analysis in the UK Biobank used a broad definition of asthma that did not consider age of onset or other potentially confounding diagnoses (described in the Methods, Pages 19-20, Lines 422-431). By comparison, Pividori et al. first excluded from consideration subjects of non-European ancestry, individuals who were related, and individuals with genetically-defined ambiguous sex assignments. From the remaining pool of 376,358 unrelated UK Biobank subjects, these investigators used age of onset of self-reported doctor-diagnosed asthma and the presence of either allergic disease or chronic obstructive pulmonary disease (COPD), emphysema, or chronic bronchitis to define childhood and adult onset cases. These factors account for most of differences between the number of asthma cases we defined and those by Pividori et al. Therefore, as suggested by the Reviewer, we carried out sensitivity analyses using the same stringent definitions as Pividori et al. by excluding non-European ancestry subjects or those individuals with allergic disease or COPD, emphysema, or chronic bronchitis. As shown in a revised Supplemental Table 9, the association results in these sensitivity analyses for the 66 novel loci were comparable to those from our primary GWAS analysis in the UK Biobank. This information is also presented in the Methods section (Pages 23-24, Lines 512-536).

9. Comment: Line 31: "UK Biobank" should be "the UK Biobank" (and elsewhere in the text).

Response: We now refer to UK Biobank as "the UK Biobank" throughout the text.

10. Comment: Line 37: chromatic should be chromatin?

Response: We apologize for the typographical error and have changed "chromatic" to "chromatin" (Page 2, Line 39).

11. Comment: Line 67-68: I would not make such a strong statement based on a single study (of relatively small sample size). Rather say something that only one study investigated rare variation, and that its results did not appear to indicate a large role for rare variation in asthma susceptibility.

Response: We agree with the Reviewer and have revised this sentence to indicate "an exome sequencing study did not provide strong evidence for rare variants playing a major role in asthma susceptibility" (Page 3, Lines 73-74).

12. Comment: Lines 74-79: Adding a brief sentence on the main analyses done would be helpful to frame the content of the paper for the reader (the paragraph currently summarizes the data sets used and the main findings but not really what was done).

Response: We have revised the last paragraph of the Introduction (Page 4, Lines 79-84) to indicate that we carried out a GWAS analysis for asthma in the UK Biobank, followed by a meta-analysis with GWAS results from the Trans-National Asthma Genetic Consortium (TAGC). These collective analyses revealed novel disease loci, provided important insight into the biological and sex-specific relevance of asthma-associated variants, and identified a potentially novel therapeutic target for treating asthma.

13. Comment: Line 112: "region" - from my reading of Supplemental Table 4 you tested single variants and not regions?

Response: We apologize for the confusion. On the right side of Supplemental Table 5 (formerly Supplemental Table 4), we have listed the lead SNPs at the 145 known asthma loci that were reported in previous publications. On the left side of the table, we compare these associations signals with ours. For example, we listed the lead SNP identified at each of the 145 known loci by our meta-analysis that was within 1Mb of the previously reported variant (left side of the table). Therefore, we did consider a region around previously reported variants but only provide the results for a single lead variant from our analyses. As shown in Supplemental Table 5, the lead variants we identified at some of the known loci were the same as those previously reported but this was not the case for all loci.

14. Comment: Lines 166-169: Asthma is an allergic disease (well, mostly). The terminology here is confusing: is this non-allergic asthma? Asthma with and without the presence of other allergic disease? This seems to be explained later in the methods section but including a brief summary here would be useful to the reader.

Response: We now indicate in the Results that we used both broad and strict definitions of asthma for the sensitivity analyses (Page 9, Lines 182-194), which is clarified further in the Methods section (Pages 23-24, Lines 512-536).

15. Comment: Figure 4 and Supplementary Table 8: enrichment for epithelium and thymus is not immediately evident to me from Figure 4 or Supplementary Table 8. In addition, Supplementary Table 8 contains 8,000 rows. I suggest submitting this instead as a supplementary data file.

Response: We have now put colored boxes in Figure 4 that match the color of the text for the tissues we highlight, including blood (red box), epithelium (orange box), lung (blue box), and thymus (purple box). In addition, we prefer to include all of the results from the GARFIELD analyses in Supplemental Table 12 (formerly Supplemental Table 8) to provide the most

information for readers, given this is only a supplementary file. Alternatively, we can include this table as part of the Journal's requirement for a source data file.

16. Comment: Lines 198-199: For reproducibility, it is important to state which public eQTL databases were used (e.g. list them in a supplementary table referred to from here).

Response: The source or reference for the public eQTL databases we used is now provided on Page 10, Lines 223-226 and in Supplemental Table 15.

17. Comment: Lines 204-205: "blood as playing an important role in asthma" - please rephrase - it is probably immune cells in blood that play a role in asthma, not blood itself!

Response: We thank the Reviewer for this suggestion, which has been made on Page 11, Line 230.

17. Comment: Line 255: "in the" should be "the"

Response: The sentence containing this typographical error has been revised (Page 13, Lines 277-279).

18. Comment: Lines 265-267: references to support these statements?

Response: We have slightly revised this sentence (Page 13, Line 285-287) and referenced a recent publication where only ~10% of loci identified for lung function traits were associated with asthma (PMID: 30804560).

19. Comment: Line 345-346: " it is possible that the genetic association results may not be generalizable to other populations" - I would rephrase this as some but not all genetic association results may be relevant in non-European populations [PMID: 30787307]

Response: We agree with the Reviewer and have modified the text to state that "...some, but not all of the genetic associations we identified may be relevant in non-European populations" and cited reference PMID: 30787307 (Page 17, Lines 377-380).

20. Comment: Figure 5, lines 513-519: How many mice were included in the experiment?

Response: We have now provided sample sizes for the mouse experiments in the text of the Legends to Figure 5 and Supplemental Figures 4 and 5.

21. Comment: Supplementary Figure 2: Please add the inflation factor at the very least to the figure legend text.

Response: We have added the inflation factor and LD Score intercept obtained from the meta-analysis to the Legend of Supplementary Figure 2. This information is also provided in the Methods (Page 21, Lines 461-463).

Response to Reviewer 3's Comments

We would like to thank the Reviewer for taking the time to review our manuscript and providing helpful critique/comments. We have addressed all the points raised and hope these revisions satisfactorily answer his/her concerns. All changes to the manuscript are indicated by blue font and Page and Line numbers.

1. Comment: This study identifies a potential lead in asthma treatment, however, the *in vivo* mechanistic insight for how CD52 may be participating in immune priming or allergic airway disease is quite limited. For instance, which mouse/human immune cells express CD52 in the disease setting? Are there particular subsets that would have relevance to allergic disease?

Response: We appreciate the Reviewer raising point. It has been shown in humans that CD52 is expressed highest on T and B lymphocytes, to a lesser extent on eosinophils and monocytes, and at very low levels or absent on neutrophils, natural killer cells, and hematopoietic stem cells (PMID: 26204829). However, we are not aware of any studies evaluating the frequency and/or function of circulating or pulmonary CD52-expressing leukocytes in human asthmatics or in mouse models of airway hyperreactivity (AHR). In this regard, our *in vivo* experiments do provide evidence that CD52-expressing B and T cells and eosinophils are recruited to the spleen and lung of allergen-exposed mice since the α CD52 antibody effectively depleted these leukocytes and diminished the induction of AHR. Despite these observations, we did not evaluate other cell populations, such as mast cells and macrophages, that could also be biologically relevant for other non-allergic sub-types of asthma. We view these as important questions that will need to be addressed in future studies.

2. Comment: The authors mention antibody staining in the methods but this is not shown.

Response: We apologize for this omission. The gating strategy used to quantitate immune cells in lung, spleen, and bronchial alveolar lavage (BAL) by flow cytometry after antibody staining is now shown in a new Supplemental Figure 4.

3. Comment: In the HDM system, type 2 cytokines contribute to both innate and adaptive phases of the response, but these are not evaluated.

Response: As part of the functional studies with the HDM system and the mouse α CD52 antibody, we elected to focus on physiological measures of pulmonary function most relevant to asthma in humans, such as lung resistance and dynamic compliance. Secondary traits of interest included cellular analysis of BAL and histological analysis of pulmonary inflammation and airway epithelium thickness. While the Reviewer is correct that type 2 cytokines can contribute to both innate and adaptive immune responses, it was not possible to measure these inflammatory markers since all of the lung tissue and aliquots of BAL were used for the analyses described above.

4. Comment: How are the human mutations that the authors connect genetically to disease contributing to function?

Response: As part of prioritizing positional candidate genes at the 66 novel loci, we leveraged publicly available expression quantitative trait locus (eQTL) data from various publicly available sources, including the GTEx Project and the eQTLGen Consortium. As described in the manuscript (Pages 10-11, Lines 223-248) and shown in Supplemental Table 15, the lead SNP or tightly linked ($r^2 > 0.8$) proxy variants at 52 of the novel loci yielded at least one *cis* eQTL for a positional candidate gene in one or more tissues, including those relevant to asthma. Notably, the most significant eQTLs were in blood and consistent with the GARFIELD bioinformatic analyses pointing to circulating immune cells in blood as playing an important role in asthma (Figure 4). eQTLs were also observed in the lung for *CD52* and *CLUAP1*, making these even stronger candidate causal genes for functional validation, of which we elected to pursue *CD52*.

Thus, evidence for how the genetic variant at each identified locus is functionally contributing to risk of asthma is based primarily on differences in mRNA levels of positional candidate genes as a function of genotype of the lead SNP. However, at some loci, the molecular mechanism for the association with asthma could be due to the variant(s) altering protein structure/function. For example, as described on Pages 14-15, Lines 318-324, the lead SNP at the chromosome 1p36.22 locus (rs2230624) is a computationally predicted deleterious Cys273Tyr substitution in *TNFRSF8*, which encodes the costimulatory molecule CD30. This member of the TNF α receptor superfamily increases pro-inflammatory cytokine production by CD4⁺ T cells and is involved in Th1, Th2, and Th17 immune responses.

5. Comment: Depletion of several immune cell types suggest that this effect is broadly immunosuppressive, consistent with prior human studies and the reduction in immune parameters.

Response: The Reviewer is correct that the mouse α CD52 antibody led to depletion of several populations of CD45⁺ cells, in particular CD4⁺ and CD8⁺ T cells and CD19⁺ B cells. These effects were observed in the spleen, which in mice is considered as accurately reflecting the pool of peripheral immune cells, and consistent with the immunosuppressive activity of alemtuzumab on the same populations of circulating lymphocytes in humans. However, by also depleting leukocytes in lung and BAL, as well as significantly reducing pulmonary inflammation, airway epithelium thickness, and allergen-induced airway hyperreactivity, the mouse α CD52 antibody had biological effects at the tissue level that, to our knowledge, have not been reported in humans. This point has been emphasized in the Discussion section (Pages 15-16, Lines 338-352).

Reviewers' Comments:

Reviewer #1:

Remarks to the Author:

All my comments have been satisfactorily addressed. I have no further comments.

Reviewer #2:

Remarks to the Author:

The authors have addressed my (reviewer 2) comments satisfactorily. W.r.t comment 7, LD score regression assumes a homogenous population, and as this was applied to a mixed cohort, the LD score regression results should at least be disclaimed as potentially being invalid (but given that this is a mostly European ancestry cohort, I do think these results may be insensitive to the violation of the assumption and should be included).

Reviewer #3:

Remarks to the Author:

The authors provide additional GWAS analysis and a thoughtful response to critiques, but unfortunately offer no new data pertaining to how the newly identified CD52 mutations might specifically impact type 2 immunity. The mouse experiments indicate that anti-CD52 depletes most peripheral T and B cell populations in addition to myeloid cells, consistent with reduced immune cell recruitment and lung responses to HDM. This broad immunodepletion strategy would of course be expected to impact experimental airway disease in mice (which is well-established to rely on lymphocytes), but also many other similar immunological challenges, consistent with use of alemtuzumab to broadly target lymphocytes in disparate immune-mediated disease settings like MS and CLL.

Response to Reviewer 2's Comments

We thank the Reviewer again for taking the time to re-review our manuscript and his/her insightful comments.

1. Comment: The authors have addressed my (reviewer 2) comments satisfactorily. W.r.t comment 7, LD score regression assumes a homogenous population, and as this was applied to a mixed cohort, the LD score regression results should at least be disclaimed as potentially being invalid (but given that this is a mostly European ancestry cohort, I do think these results may be insensitive to the violation of the assumption and should be included).

Response: In the Methods, we now acknowledge that while LD Score regression assumes a homogenous population, it could still be applicable to the UK Biobank since most subjects are of European ancestry (Page 21, Lines 442-444).

Response to Reviewer 3's Comments

We thank the Reviewer again for taking the time to re-review our manuscript and his/her insightful comments.

1. Comment: The authors provide additional GWAS analysis and a thoughtful response to critiques, but unfortunately offer no new data pertaining to how the newly identified *CD52* mutations might specifically impact type 2 immunity. The mouse experiments indicate that anti-*CD52* depletes most peripheral T and B cell populations in addition to myeloid cells, consistent with reduced immune cell recruitment and lung responses to HDM. This broad immunodepletion strategy would of course be expected to impact experimental airway disease in mice (which is well-established to rely on lymphocytes), but also many other similar immunological challenges, consistent with use of alemtuzumab to broadly target lymphocytes in disparate immune-mediated disease settings like MS and CLL.

Response: We appreciate the Reviewer's concern regarding this point. The *CD52* locus was identified in a GWAS analysis for a broadly defined asthma phenotype and not for any specific sub-type. In attempting to validate *CD52* as at least one candidate gene at this locus, we chose a mouse model of allergen-induced model of airway hyperreactivity (AHR) simply as a proof of concept. While we did observe attenuation of HDM-induced AHR by the anti-*CD52* antibody, our intention was not to claim that *CD52* specifically impacted type 2 immunity. This question could be addressed, for example, by testing the association of the *CD52* locus with various sub-types of asthma in humans or by determining whether the anti-*CD52* antibody affects induction of AHR by other stimuli that do not involve a type 2 immune response. We believe these are important questions to address in future studies. As a result, and in response to the Reviewer's and Editor's concerns, we have raised this point in the Discussion and removed portions of the text so as to temper the conclusions of the functional mouse experiments with respect to any translational implications or clinical utility of an anti-*CD52* antibody for asthma in humans (Pages 17, Lines 360-363).